# Structural mechanism of voltage-gated sodium channel slow inactivation

Huiwen Chen[1,2,7], Zhanyi Xia[2,3,7], Jie Dong[4,7], Bo Huang [5,7], Jiangtao Zhang[2,6], Feng Zhou[5], Rui Yan[2,6], Yiqiang Shi[4], Jianke Gong [6], Juquan Jiang [1] ✉, Zhuo Huang [4] ✉ & Daohua Jiang [2,3] ✉

Voltage-gated sodium (Na$_V$) channels mediate a plethora of electrical activities. Na$_V$ channels govern cellular excitability in response to depolarizing stimuli. Inactivation is an intrinsic property of Na$_V$ channels that regulates cellular excitability by controlling the channel availability. The fast inactivation, mediated by the Ile-Phe-Met (IFM) motif and the N-terminal helix (N-helix), has been well-characterized. However, the molecular mechanism underlying Na$_V$ channel slow inactivation remains elusive. Here, we demonstrate that the removal of the N-helix of Na$_V$Eh (Na$_V$Eh$^{\Delta N}$) results in a slow-inactivated channel, and present cryo-EM structure of Na$_V$Eh$^{\Delta N}$ in a potential slow-inactivated state. The structure features a closed activation gate and a dilated selectivity filter (SF), indicating that the upper SF and the inner gate could serve as a gate for slow inactivation. In comparison to the Na$_V$Eh structure, Na$_V$Eh$^{\Delta N}$ undergoes marked conformational shifts on the intracellular side. Together, our results provide important mechanistic insights into Na$_V$ channel slow inactivation.

Voltage-gated sodium (Na$_V$) channels are responsible for the initiation and propagation of electrical signals in nerves, muscles, cardiomyocytes, and other excitable cells[1–3]. Vertebrate Na$_V$ channels are composed of a large pore-forming α-subunit with over 2000 residues and one or two small auxiliary β-subunits[4,5]. The ancestral Na$_V$ channels for the asymmetric four-domain vertebrate Na$_V$ channels have been identified in bacteria[6] and eukaryotic unicellular phytoplankton[7], which are organized in a homo-tetrameric fashion. Despite the structural diversity, Na$_V$ channels share a similar activation process, that can be simplified into three states: resting state, activated open state in response to membrane depolarization, and inactivated state[3,8]. Inactivation of the Na$_V$ channel terminates sodium influx and thus determines the availability of the channel. It has been implied that the

inactivation of the Na$_V$ channel is composed of a fast (a few ms) component, a slow (tens of hundreds of ms) component, and even an ultra-slow component (tens of sec to minutes)[9–11]. Fast inactivation is the hallmark feature of eukaryotic Na$_V$ channels. Extensive functional and structural studies have elucidated the molecular mechanism of Na$_V$ channel fast inactivation, that is, a triple hydrophobic residue motif Ile-Phe-Met (IFM) located in the intracellular loop between domain III (DIII) and DIV serves as a hydrophobic latch closing the activation gate[5,12–15]. Interestingly, we have recently described an unexpected N-terminal helix (N-helix) mediated N-type fast inactivation of Na$_V$Eh from the coccolithophore *Emiliania huxleyi*[16], which is mechanistically distinct from the IFM-motif mediated fast inactivation but similar to the ball-and-chain inactivation of potassium channels[17].

[1]Department of Microbiology and Biotechnology, College of Life Sciences, Northeast Agricultural University, No. 600 Changjiang Road Xiangfang District Harbin 150030, China. [2]Laboratory of Soft Matter Physics, Institute of Physics, Chinese Academy of Sciences, Beijing 100190, China. [3]School of Physical Sciences, University of Chinese Academy of Sciences, Beijing 100190, China. [4]State Key Laboratory of Natural and Biomimetic Drugs, Department of Molecular and Cellular Pharmacology, School of Pharmaceutical Sciences, Peking University Health Science Center, Beijing 100191, China. [5]Beijing Stone-Wise Technology Co Ltd., 15 Haidian street Haidian district Beijing, China. [6]College of Life Science and Technology, Key Laboratory of Molecular Biophysics of MOE, Huazhong University of Science and Technology, Wuhan, Hubei, China. [7]These authors contributed equally: Huiwen Chen, Zhanyi Xia, Jie Dong, Bo Huang. ✉e-mail: jjqdainty@163.com; huangz@hsc.pku.edu.cn; jiangdh@iphy.ac.cn

Prolonged depolarization or repetitive depolarizing pulses over a duration of tens of seconds cause slow inactivation of Na$_V$ channel, a phenomenon that is significantly different from the fast inactivation in the context of onset and recovery rate[18,19]. Slow inactivation reduces the number of conductive channels, regulates excitability, and protects cells from aberrant high-repetitive stimuli that can lead to excitotoxicity[11,20–22]. Meanwhile, pathogenic mutations that impair slow inactivation have been reported to be associated with human diseases including hyperkalemic periodic paralysis, myotonia, and Brugada syndrome[11,23–26]. Despite a large body of evidence indicating that the voltage-sensor domain (VSD), pore-loop region, and the fast inactivation gate are involved in the process of slow inactivation[10,11], the molecular mechanism of this inherent property of Na$_V$ channels remains unclear. Because of lacking fast inactivation, prokaryotic Na$_V$ channels have been used as models to study the slow inactivation[27–31]. Interestingly, the crystal structure of wild-type Na$_V$Ab (Na$_V$Ab$^{WT}$) adopts a twofold symmetric configuration with both collapsed selectivity filter (SF) and intracellular activation gate[29], revealing a possible slow-inactivated state which is in agreement with previous reports showing that asymmetric deformation and conformational changes of the pore domain are involved in slow inactivation[27,29,30]. Although the structure provided a plausible possible explanation for Na$_V$ channel slow inactivation, however, the conformations could be affected by protein packing in the crystal lattices and crystallization conditions that are remarkably different from physiological conditions[29]. In addition, a lipid-infiltration model was proposed to explain slow inactivation[28]. Lipid molecules and local anesthetic drugs are thought to act as reversible inhibitors, which slowly penetrate the fenestrations into the pore and block the channel[28,32,33]. However, lipid molecules were not consistently found within the fenestrations of Na$_V$ channels[16,27]. Taken as a whole, none of these proposed mechanisms can satisfactorily explain the phenomenon of slow inactivation. The structural mechanism of Na$_V$ channel slow inactivation remains incomplete and controversial.

In this work, we first precluded the possible Ca$^{2+}$-dependent activation of the wild-type Na$_V$Eh (Na$_V$Eh$^{WT}$)[7], and measured the slow inactivation properties of the N-helix truncated Na$_V$Eh (Na$_V$Eh$^{ΔN}$). We next determine the cryo-electron microscopy (cryo-EM) structure of Na$_V$Eh$^{ΔN}$, which displays unexpected conformational shifts in both the intracellular activation gate and the extracellular selectivity filter in comparison to the Na$_V$Eh$^{WT}$ structure. These results indicate that the Na$_V$Eh$^{ΔN}$ structure is captured in a possible slow-inactivated state. Collectively, these findings provide valuable insights into the molecular mechanism of the Na$_V$ channel slow inactivation.

## Results

### Functional characterization of Na$_V$Eh$^{ΔN}$

Activation of Na$_V$Eh was previously suggested to be dependent on extracellular Ca$^{2+}$ (ref. 7). To assess the potential Ca$^{2+}$-dependent activation, we measured Na$_V$Eh$^{WT}$ currents in the presence of Ca$^{2+}$ or EGTA using the whole-cell patch clamp recording of HEK293 cells. The Na$_V$Eh-expressing cells consistently generated robust inward currents regardless of with Ca$^{2+}$ or EGTA (Supplementary Fig. 1a and 1b). To further assess the potential regulation of Na$_V$Eh by Ca$^{2+}$ from a structural aspect, we determined the cryo-EM structures of Na$_V$Eh in the presence of 2 mM Ca$^{2+}$ (Na$_V$Eh$^{WT\_Ca}$) and 2 mM EGTA (Na$_V$Eh$^{WT\_EGTA}$), respectively (Supplementary Fig. 2 and Supplementary Table 1). The resulting two structures are essentially identical with a root mean square deviation (RMSD) of 0.37 Å, and only subtle conformational shifts between the side-chains in the extracellular loops (ECLs) of the structures can be observed (Supplementary Fig. 1c, d). These results confirm that Ca$^{2+}$ has negligible effects on the function and structure of Na$_V$Eh.

After excluding Ca$^{2+}$ regulation of Na$_V$Eh, we next focused on the slow inactivation of Na$_V$Eh. The deletion of the N-terminal Ile2-Arg13

drastically impaired the fast inactivation of Na$_V$Eh[16], which is consistent with the loss of fast inactivation when substitution of the IFM-motif with three glutamine residues in mammalian Na$_V$1.5 (Na$_V$1.5/QQQ)[14] and the deletion of the N-terminus of Shaker potassium channel[34]. We further investigated the functional properties of the N-terminal Ile2-Arg13 removed Na$_V$Eh (Na$_V$Eh$^{ΔN}$). The Na$_V$Eh$^{ΔN}$-expressing cells exhibited rapid activation in response to a train of depolarizing stimuli, the amplitude of which slowly decreased, in sharp contrast to the fast inactivation of Na$_V$Eh$^{WT}$ (Fig. 1a, b). Notably, the inactivation of Na$_V$Eh$^{ΔN}$ appears to be composed of two components, a relatively fast component within the first 10 ms and a prolonged slow reduction of the current. To assess the two components of the inactivation, we generated a series of Na$_V$Eh mutants with longer N-terminal deletions, including Na$_V$Eh$^{ΔN21}$ (deletion of Ile2-Ala21), Na$_V$Eh$^{ΔN48}$ (deletion of Ile2-Ala48), and Na$_V$Eh$^{ΔN59}$ (deletion of Ile2-Leu59). The resulting mutants turned out to be non-functional (Supplementary Fig. 3a–g). Alternatively, we mutated $^{15}$AAAA$^{18}$ of Na$_V$Eh$^{ΔN}$ to $^{15}$EEEE$^{18}$, which would increase the repulsion of the N-terminal loop with the negatively charged outer mouth of the activation gate[16]. The resulting construct Na$_V$Eh$^{ΔN18E}$ exhibited almost complete ablation of fast inactivation (Supplementary Fig. 3a, d). These observations resemble the fact that substitution of the IFM-motif with QQQ almost fully abolished fast inactivation of mammalian Na$_V$ channels, while the mutants with single-site mutations of F1489Q in brain Nav channel and F1485Q in heart Nav1.5 exhibited 15% and 70% residual fast inactivation[12,35], respectively. These results indicate that the residual fast inactivation of Na$_V$Eh$^{ΔN}$ is still mediated by the residual N-terminal fast inactivation particle.

The current density of Na$_V$Eh$^{ΔN}$ is only one-ninth of Na$_V$Eh$^{WT}$ (Fig. 1e), which is in line with the observation that the peak current of Na$_V$1.5 F1485Q mutant is one-fifth of WT Na$_V$1.5[35]. In addition, the voltage-dependence of activation is almost identical to that of Na$_V$Eh$^{WT}$ (ref. 16), yielding a $V_{1/2}$ value of −60.7 ± 1.1 mV (Fig. 1f, $n = 5$). Compared to Na$_V$Eh$^{WT}$, the voltage-dependence of inactivation of Na$_V$Eh$^{ΔN}$ is shifted toward positive potential by ~15.5 mV. Similar unchanged voltage-dependence of activation and positive shift of voltage-dependence of inactivation were observed in human Na$_V$1.5 with the F1485Q mutation[35]. Furthermore, Na$_V$Eh$^{ΔN}$ showed incomplete inactivation with ~30% resistance under the conditioning pulse of 30 s (Fig. 1c, f), which was also observed in the Na$_V$1.5 F1485Q mutant[35]. Because the binding of the fast inactivation particles stabilizes the activation gates[14–16] and the removal of the fast inactivation particle enhances slow inactivation[36–38], the incomplete inactivation of Na$_V$Eh$^{ΔN}$ and Na$_V$1.5$^{F1485Q}$ could be caused by the residual fast inactivation particles interacting with the activation gate. We further sought to measure the recovery rate of Na$_V$Eh$^{ΔN}$ from inactivation. After a long pre-pulse of 30 s, the currents were gradually recovered when the cells were held at −150 mV with elongated time up to 33 s (Fig. 1d). The resulting recovery curve shows two kinetic components of $\tau_{fast} = 20.1 ± 3.7$ ms and $\tau_{slow} = 8714 ± 2505$ ms (Fig. 1g), presumably corresponding to the recovery from the residual fast inactivation and slow inactivation, respectively. These results demonstrate that Na$_V$Eh$^{ΔN}$ undergoes a slow inactivation process, which resembles the slow inactivation observed in other Na$_V$ channels[6,35,39], suggesting that Na$_V$Eh$^{ΔN}$ is suitable for structural analysis of slow inactivation of Na$_V$ channels.

### Cryo-EM structure of Na$_V$Eh$^{ΔN}$

To reveal the structural basis of Na$_V$Eh$^{ΔN}$ slow inactivation, we sought to determine the cryo-EM structure of Na$_V$Eh$^{ΔN}$. Unlike the over-expression of Na$_V$1.5/QQQ in HEK293F cells causing cytotoxicity[14], Na$_V$Eh$^{ΔN}$ expressing HEK293F cells appeared to be healthy, presumably because of the reduced current density and the inactivation of Na$_V$Eh$^{ΔN}$ (Fig. 1c). The subsequent purification of Na$_V$Eh$^{ΔN}$ yielded homogeneous samples for cryo-EM analysis (Supplementary Fig. 4). To validate

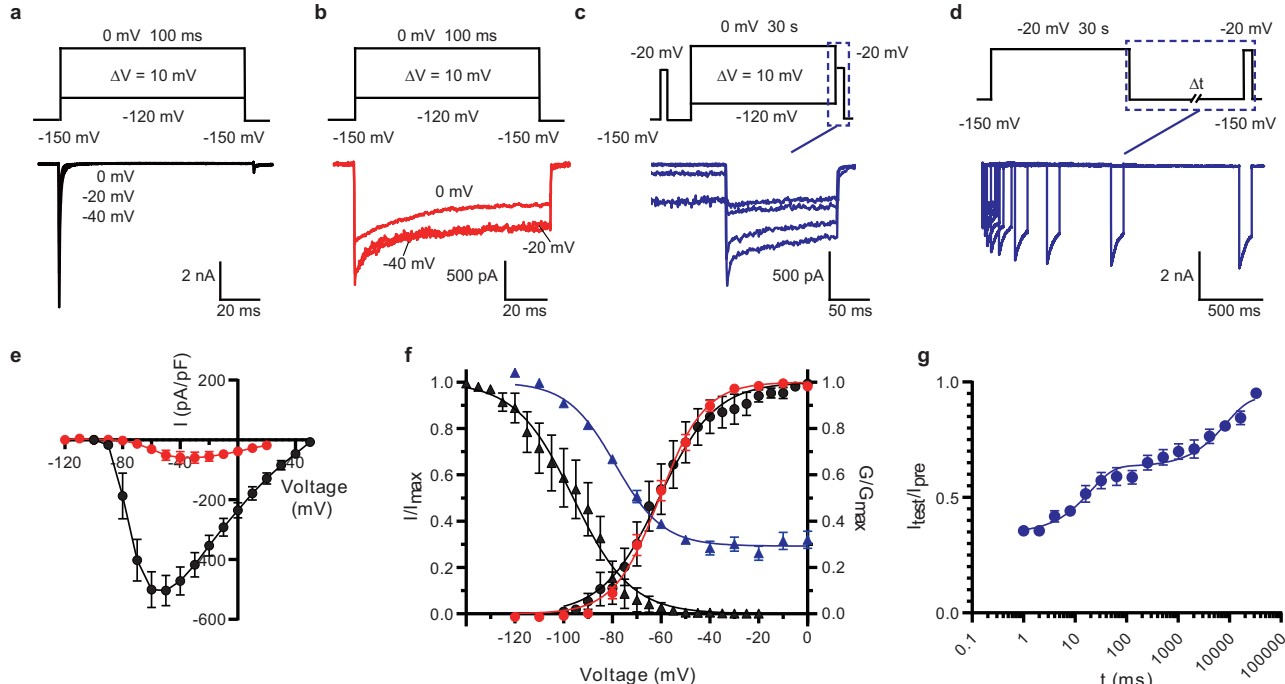

**Fig. 1 | Functional characteristics of Na$_V$Eh$^{\Delta N}$. a–c** Representative current traces of Na$_V$Eh$^{WT}$ activation (**a**), Na$_V$Eh$^{\Delta N}$ activation (**b**), and Na$_V$Eh$^{\Delta N}$ inactivation (**c**). A schematic diagram of the recording protocol is presented on top of the respective current traces. **d** Representative current traces of Na$_V$Eh$^{\Delta N}$ recovery from inactivation. Currents were elicited by a pre-pulse at −20 mV for 30 s, followed by an inter-pulse ranging from 1 ms to 33 s at −150 mV. After inter-pulse, a 100 ms test pulse at −20 mV was applied. A schematic diagram of the recording protocol is presented on the top of each current trace. **e** I–V relationship for Na$_V$Eh$^{\Delta N}$ (Red, $n = 5$) and Na$_V$Eh$^{WT}$ (Black, $n = 9$). Currents were normalized to cell capacitance. **f** Normalized conductance-voltage (G/V) relationship and steady-state inactivation of Na$_V$Eh$^{\Delta N}$ (Red and Blue) and Na$_V$Eh$^{WT}$ (Black). For measuring G/V curve, currents were elicited by 100 ms depolarizing pulses between −120 mV and 0 mV in step of 10 mV from a holding potential of −150 mV. For measuring steady-state inactivation of Na$_V$Eh$^{\Delta N}$ (Blue), currents were elicited by a pre-pulse between −120 mV and 0 mV in 10 mV increments for 30 s and followed by a 100 ms test pulse at −20 mV. The Boltzmann fitted data yielded Na$_V$Eh$^{\Delta N}$ activation (Red) $V_{1/2} = -60.7 \pm 1.1$ mV ($n = 5$) and steady-state inactivation (Blue) $V_{1/2} = -78.9 \pm 2.0$ mV ($n = 5$). The Na$_V$Eh$^{WT}$ (Black) activation $V_{1/2} = -61.5 \pm 2.1$ mV ($n = 15$) and steady-state fast inactivation $V_{1/2} = -94.4 \pm 2.1$ mV ($n = 9$) are adapted from our previous study. **g** The time course for recovery from steady-state slow inactivation of Na$_V$Eh$^{\Delta N}$. Recovery curve from slow inactivation was fitted using a double exponential function which yielded two kinetic components $\tau_{fast} = 20.1 \pm 3.7$ ms, $\tau_{slow} = 8714 \pm 2505$ ms ($n = 5$). Data are means ± SEM, $n$ = the number of different cells. Source data are provided as a Source Data file.

whether any potential symmetry differences in Na$_V$Eh$^{\Delta N}$, 3D classifications were performed with C1-symmetry imposed, all the resulting 3D maps with discernible densities for the transmembrane helices display apparent fourfold symmetry (Supplementary Fig. 5a). The final EM maps of Na$_V$Eh$^{\Delta N}$ were refined to nominal resolutions of 3.5 Å with C1-symmetry and 3.1 Å with C4-symmetry imposed, respectively (Supplementary Fig. 5a–d). Notably, the two EM maps rich in side-chain details are nearly identical, strongly indicating that the Na$_V$Eh$^{\Delta N}$ structure adopts the fourfold symmetric organization despite the lack of the N-helix. The high-quality EM map allows reliable model building of Na$_V$Eh$^{\Delta N}$ and the assignment of possible lipids (Fig. 2a, b, Supplementary Fig. 5e).

The overall structure of Na$_V$Eh$^{\Delta N}$ is rather similar to Na$_V$Eh$^{WT}$ (with an RMSD of 2.47 Å), both of which are organized in a domain-swapped manner and are surrounded by lipid molecules (Supplementary Fig. 6a–f). However, substantial local conformational differences in the ECL, SF, and activation gate can be observed between the two structures. The ECL region (Asp242-Gly280) of Na$_V$Eh$^{\Delta N}$ is invisible because of the conformational changes in the P2 helix (Fig. 2a, b, Supplementary Fig. 6a, b). Interestingly, the fenestrations of Na$_V$Eh$^{\Delta N}$ were observed to be smaller than that of Na$_V$Eh$^{WT}$ because of the side-chain rotation of F218 and Y321 (Supplementary Fig. 6g, h), in line with previous studies showing that the fenestrations of Na$_V$ channel could undergo conformational changes during state-transitions and may serve as gates for state-dependent inhibitors[33,40,41].

It is worth noting that the activation gate of Na$_V$Eh$^{\Delta N}$ appears to be dramatically smaller than that of Na$_V$Eh$^{WT}$, lacking discernible EM density inside it; by contrast, the larger activation gate of Na$_V$Eh$^{WT}$ is blocked by its N-helix (Fig. 2c). More precisely, the Van der Waals diameter for the activation gate of Na$_V$Eh$^{\Delta N}$ is less than 2 Å (Fig. 2d–f), which is much smaller than the required diameter of ~7.5 Å for conducting hydrated Na$^+$ (refs. 16,42), indicating that the activation gate of Na$_V$Eh$^{\Delta N}$ is non-conductive. These structural observations suggest that Na$_V$Eh$^{\Delta N}$ was captured in a possible slow-inactivated state that is distinct from the Na$_V$Eh$^{WT}$ in the fast-inactivated state mediated by the N-helix[16].

### Structural basis of Na$_V$Eh$^{\Delta N}$ slow inactivation
To define the molecular determinants for the slow inactivation of Na$_V$Eh$^{\Delta N}$, we carefully compared the structures of Na$_V$Eh$^{\Delta N}$ and Na$_V$Eh$^{WT}$. The superposition of Na$_V$Eh$^{\Delta N}$ and Na$_V$Eh$^{WT}$ demonstrates that the size of the channel at the extracellular side remains unchanged, whereas the intracellular side of Na$_V$Eh$^{\Delta N}$ is shrunken by ~8 Å in comparison to Na$_V$Eh$^{WT}$ (Fig. 3a). The VSDs of the two structures assume activated conformation and are essentially identical (with an RMSD of 0.5 Å), in line with the same purification conditions of no membrane potential and their unchanged voltage-dependence of activation (Figs. 1d and 3c). However, the VSDs of Na$_V$Eh$^{\Delta N}$ were rotated by ~10 degrees towards the center of the activation gate, which in turn shifts the S4-S5 linker helices and the pore-lining S6 helices (Fig. 3b). As a result, the activation gate of Na$_V$Eh$^{\Delta N}$ is almost fully-closed by the four L350 residues, which is apparently much smaller than the open gate of Na$_V$Eh$^{WT}$ (Fig. 3d). In addition, the activation gate of Na$_V$Eh$^{\Delta N}$ appears to be different from that of the resting Na$_V$Ab (PDB code: 6P6W[43]), the

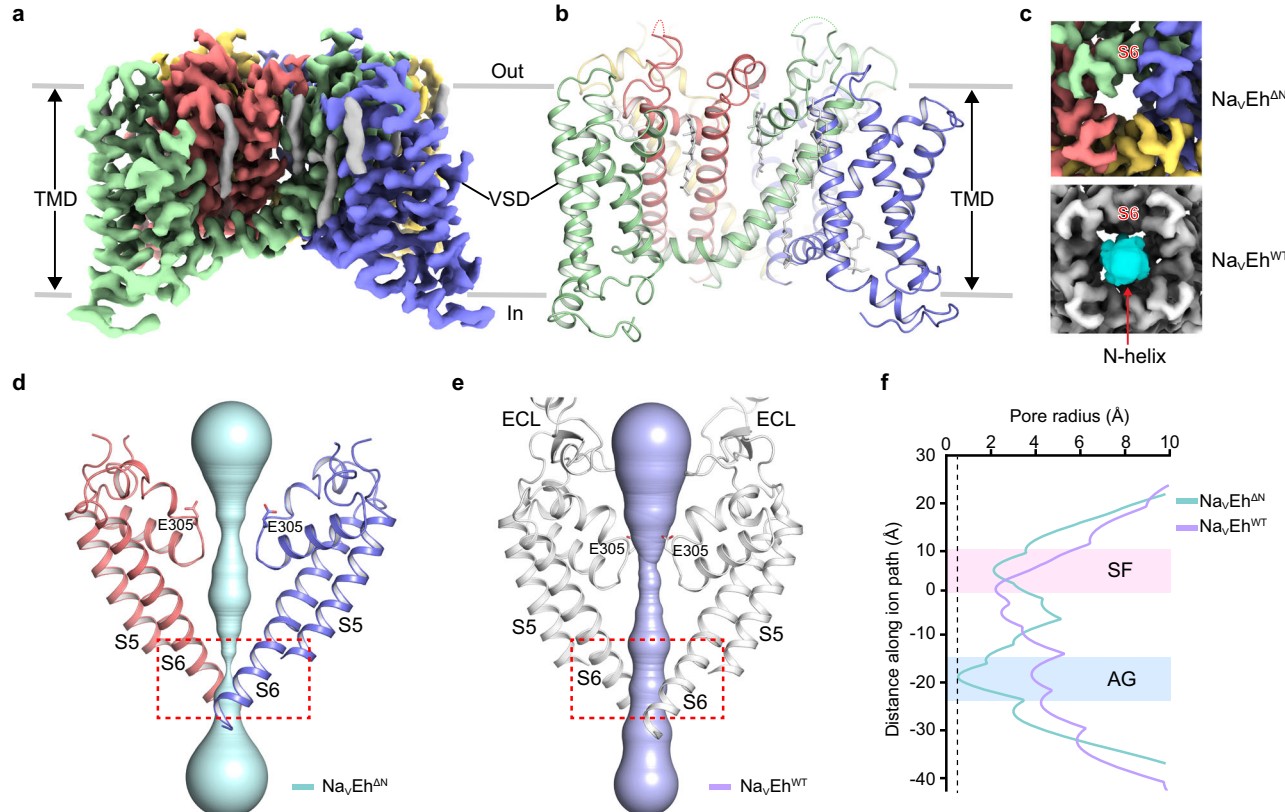

**Fig. 2 | Cryo-EM structure of Na$_V$Eh$^{\Delta N}$. a** The cryo-EM map of Na$_V$Eh$^{\Delta N}$. The four subunits and lipids are colored in green, light purple, yellow, red, and gray, respectively. The same color codes are applied for Na$_V$Eh$^{\Delta N}$ unless specified. **b** Cartoon representation of Na$_V$Eh$^{\Delta N}$. **c** The activation gate of Na$_V$Eh$^{\Delta N}$ (upper panel) and Na$_V$Eh$^{WT}$ (lower panel). The Na$_V$Eh$^{WT}$ map is colored in gray, and the N-helix is highlighted in cyan. **d**, **e** Ion path of Na$_V$Eh$^{\Delta N}$ (**d**) and Na$_V$Eh$^{WT}$ (**e**) calculated by HOLE. Residues at the constriction sites of selectivity filter are shown side chains in sticks. The red dashed boxes indicate the regions of intracellular activation gates. **f** Pore radii of Na$_V$Eh$^{\Delta N}$ and Na$_V$Eh$^{WT}$ from **d**, **e**. The pore radii of Na$_V$Eh$^{\Delta N}$ and Na$_V$Eh$^{WT}$ are colored cyan and light purple, respectively.

pre-open Na$_V$Ab (PDB code: 3RVY[27]), the open Na$_V$Ms (PDB code: 5HVX[44]), and the fast-inactivated Na$_V$1.5 (PDB code: 6UZ0[15]) (Supplementary Fig. 7), suggesting that the activation gates of Na$_V$ channels can be finely regulated.

Unexpectedly, unambiguous EM densities reveal that the P2 helix in the pore-loop region of Na$_V$Eh$^{\Delta N}$ adopts a different conformation in comparison to that of the Na$_V$Eh$^{WT}$ (Supplementary Fig. 5f). The three-turn α helix of P2 was broken into two shorter P2a and P2b helices at the position of P314, of which the P2b helix underwent a displacement of ~9 Å away from the center of the SF (Fig. 3e). Because of the movement, Na$_V$Eh$^{\Delta N}$ has a dilated vestibule. Of note, this Pro in the P2 helix is conserved in most homo-tetrameric one-domain Na$_V$ channels and in DIV of the asymmetric four-domain Na$_V$ channels (Supplementary Fig. 8), suggesting that the P2 helix may undergo similar conformational changes in the pore-loop region under certain conditions. However, such conformational changes in the P2 helix were not observed in the reported Na$_V$ channel structures[13,15,29,43,44]. Convincingly, the functional and structural evidence elucidated that the SF mediates the slow inactivation (so-called C-type inactivation) of K$^+$ channels[45–47]. Given the fact that many mutagenesis studies have shown that the pore-helix regions are involved in Na$_V$ channel slow inactivation[11,30,48–50], we therefore propose that the structure of Na$_V$Eh$^{\Delta N}$ may represent an unseen slow-inactivated state of Na$_V$ channels.

### The dilated selectivity filter of Na$_V$Eh$^{\Delta N}$
Na$_V$ channels feature a short SF between P1 and P2 helices, of which the well-characterized "DEKA" locus of asymmetric eukaryotic Na$_V$ channels and the "EEEE" of fourfold symmetric Na$_V$ channels function as essential molecular determinants for Na$^+$ selectivity and conductance[27,51]. Within the SF of Na$_V$Eh$^{WT}$, four E305 residues are properly oriented to form a 'high-field strength' (HFS) site that attracts extracellular hydrated Na$^+$ (Fig. 4a, d). Strikingly, the dilated SF of Na$_V$Eh$^{\Delta N}$ was expanded by 6.7 Å between the two opposing E305 residues and by 4.8 Å between the two neighboring E305 residues in comparison to the SF of Na$_V$Eh$^{WT}$ (Fig. 4b, e). A superposition of the SFs of Na$_V$Eh$^{\Delta N}$ and Na$_V$Eh$^{WT}$ reveals that the SF expansion of Na$_V$Eh$^{\Delta N}$ is caused by the conformational changes in the P2 helix, whereas the P1 helix exhibits only minor conformational shifts (Fig. 4c). Most likely, the expanded E305 cannot form a proper HFS site for coordinating hydrated Na$^+$ (Fig. 4e). Additionally, in the middle of the SF, the carboxyl oxygen atoms of G304 guide the conductance of partially dehydrated Na$^+$ by the HFS site (Fig. 4a, d). It is noteworthy that the dimension of G304 in Na$_V$Eh$^{\Delta N}$ appears to be slightly constricted by 0.4 Å when compared to Na$_V$Eh$^{WT}$, thereby further restricting the access of extracellular fully-hydrated ions to the channel. Furthermore, the highly-conserved T303-W307 hydrogen-bond pairs, which stabilize the SFs of Na$_V$ and Ca$_V$ channels[27], seem to separate from each other and form a weak interaction at a distance of 4 Å in Na$_V$Eh$^{\Delta N}$ (Fig. 4d–f), further implying that the expanded SF of Na$_V$Eh$^{\Delta N}$ may not be suitable for conducting Na$^+$.

The previous Na$_V$Ab structure suggested that the collapse of both the SF and activation gate may be involved in the inactivation process[29]. Despite the symmetry rearrangement of the fourfold Na$_V$Ab[27] (Na$_V$Ab$^{C4}$) to the two-fold Na$_V$Ab[29] (Na$_V$Ab$^{C2}$) resulting in a non-conductive SF and activation gate (Fig. 4g, h, Supplementary Fig. 7b, c), the SF and pore helices of Na$_V$Ab$^{C2}$ adopt a nearly identical conformation to that of Na$_V$Ab$^{C4}$ (Fig. 4i). In contrast to the Na$_V$Ab$^{C2}$, Na$_V$Eh$^{\Delta N}$ maintains a four-fold symmetric configuration while exhibiting significant

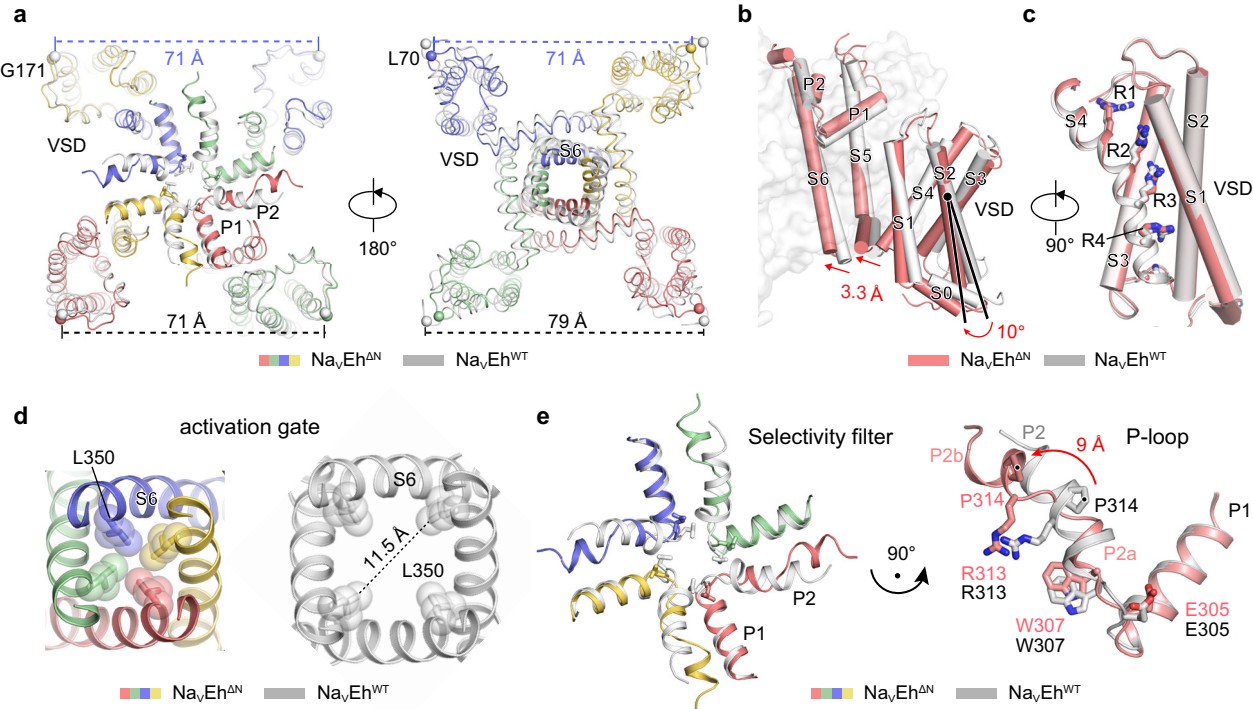

**Fig. 3 | Structural comparison of Na$_V$Eh$^{WT}$ and Na$_V$Eh$^{\Delta N}$. a** The superposition of Na$_V$Eh$^{\Delta N}$ and Na$_V$Eh$^{WT}$ (gray) viewed from the extracellular side (Left) and from the intracellular side (Right). The distances between neighboring subunits are labeled. **b** Conformational changes of Na$_V$Eh$^{\Delta N}$ (red) and Na$_V$Eh$^{WT}$ (gray). One subunit is shown in cartoon, and the other three are shown in half-transparent surface. The pore domains of two structures are superimposed. Red arrows indicate the conformational shifts. **c** The superposition of the VSDs of Na$_V$Eh$^{\Delta N}$ (red) and Na$_V$Eh$^{WT}$ (gray). S1-S3 are shown as cylindrical helices and S4 is shown as cartoon. Gating charge residues are shown as sticks. **d** The activation gate of Na$_V$Eh$^{\Delta N}$ (Left) and Na$_V$Eh$^{WT}$ (Right), respectively. The key hydrophobic residues are shown as sticks and spheres. **e** Selectivity filter comparison between Na$_V$Eh$^{\Delta N}$ and Na$_V$Eh$^{WT}$. On the right, conformational changes of the P2 helix between Na$_V$Eh$^{\Delta N}$ and Na$_V$Eh$^{WT}$. The red arrow indicates the conformational shift of P2 helix.

conformational changes in both the SF and the activation gate. These observations imply that the SF of Na$_V$ channels could potentially function as a gate for slow inactivation through distinct mechanisms.

The K$^+$ channel structures of KcsA[45] and Shaker K$_V$ channel (with a W434F mutation in the SF)[46] evidently demonstrated that the plasticity of the SFs potentially determines their slow inactivation. Similar to our Na$_V$Eh$^{\Delta N}$, the fourfold symmetric KcsA and Shaker K$_V^{W434F}$ display conformational rearrangements in their SFs that cause altered K$^+$ interactions within the SFs (Supplementary Fig. 9). Unlike Na$_V$Eh$^{\Delta N}$, the intracellular gates of KcsA and Shaker K$_V^{W434F}$ appeared to assume open conformations while the activation gate of Na$_V$Eh$^{\Delta N}$ is closed. Nevertheless, these findings elucidate that the SFs of Na$_V$ and K$^+$ channels play dual roles in ion selectivity and slow inactivation.

## MD simulations of the pore domain of Na$_V$Eh

To validate the cryo-EM structures of Na$_V$Eh$^{WT}$ and Na$_V$Eh$^{\Delta N}$, we conducted molecular dynamics (MD) simulations using the pore domains of Na$_V$Eh$^{WT}$ and Na$_V$Eh$^{\Delta N}$ as initial configurations for a duration of 2 µs (Supplementary Fig. 10a). We investigated the stability of key structural features, including the dilation of the SF, size of fenestrations, and distances between opposing helices. Specifically, we analyzed the distance between the alpha carbon (Cα) atoms of residues A310 and P314, the distance between the side-chain centroids of F299 and F218, and the pore radius as indicators of SF dilation, fenestrations, and distances between opposing helices, respectively. The consistent values of these indicators suggest the stability of these key structural features (Supplementary Fig. 10b–d). Furthermore, a clustering analysis was performed on the MD results to identify the most frequently observed conformations of the SF, which shows two major clusters (comprising 85% of the population) for the SF of Na$_V$Eh$^{\Delta N}$ and seven

major clusters (each comprising over 5% of the population) for the SF of Na$_V$Eh$^{WT}$ (Supplementary Fig. 10e).

To further dissect the relationship between activated and inactivated states of Na$_V$Eh, we performed a 4 µs MD simulation on the open pore domain of Na$_V$Eh$^{WT}$ structure by removing the N-helix. After plotting the pore radius at the activation gate of each MD simulation trajectory, the results show that the gate remained open for a duration of 300–680 ns and subsequently became closed (Supplementary Fig. 11a). Consistent with the findings of previous studies[14,52], this result indicates that the open gate of Na$_V$ channels is inherently unstable. The clustering of the MD trajectories with an RMSD cutoff of 1.5 Å resulted in 4–6 populations for each simulation, of which the majority of populations displayed closed activation gates (Supplementary Fig. 11b, c). We selected the representative trajectory of the most populated cluster of the five MD simulations and compared them with the pore domain of Na$_V$Eh$^{\Delta N}$. The superposition reveals that the MD representatives are highly similar to the EM structure with backbone RMSD of 1.5–1.8 Å, especially between the activation gates (Supplementary Fig. 11d). Notably, the P2 helix of the five MD representatives adopt an intact α-helical conformation similar to that of Na$_V$Eh$^{WT}$, unlike the broken P2 helix in Na$_V$Eh$^{\Delta N}$ (Supplementary Fig. 11e, f). This observation suggests that under our simulation conditions, it is difficult to achieve the slow-inactivated state, in line with the observation that the development of slow inactivation on the order of tens of seconds[53,54]. To tackle this limitation, we are actively exploring enhanced sampling methods, such as meta-dynamics. The second factor is the truncation of the VSD during our MD simulations. It is possible that this truncation would impair the integrity of the whole system and thereby block the desired conformational changes. Nevertheless, we anticipate that future MD simulations incorporating

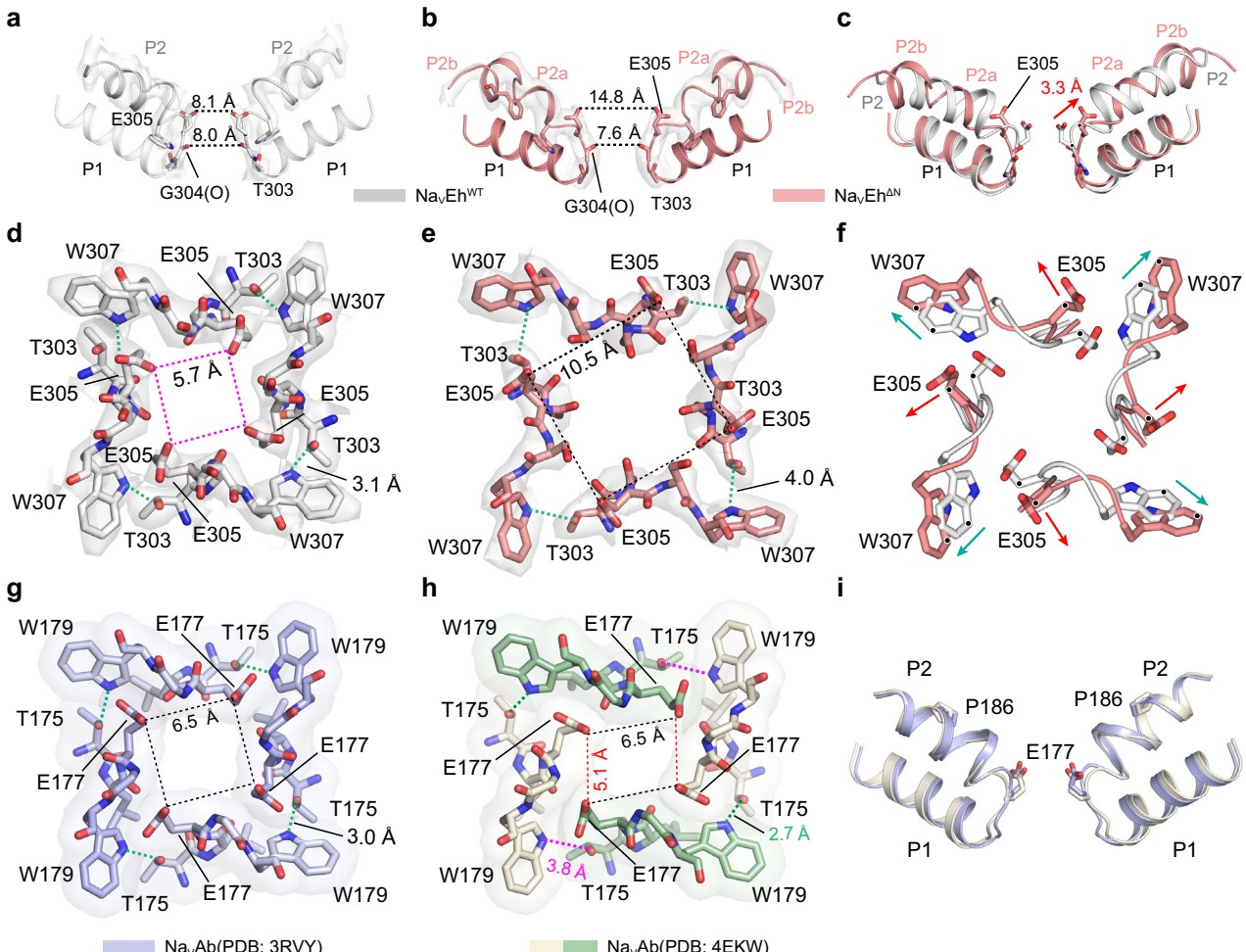

**Fig. 4 | The dilated selectivity filter of Na$_V$Eh$^{\Delta N}$. a, b** The selectivity filter of Na$_V$Eh$^{WT}$ (**a**) and Na$_V$Eh$^{\Delta N}$ (**b**). Black dashed lines represent the distances within the SFs. EM densities for P2 helices are presented in white surface contoured at 10 σ. **c** The superposition of the pore region of Na$_V$Eh$^{WT}$ (gray) and Na$_V$Eh$^{\Delta N}$ (red). E305 is shown side chain in sticks. The red arrow indicates the shifts of E305 between the two SFs. **d, e** The top view of the selectivity filter of Na$_V$Eh$^{WT}$ (**d**) and Na$_V$Eh$^{\Delta N}$ (**e**). The red and black dashed square represents the SF size of Na$_V$Eh$^{WT}$ and Na$_V$Eh$^{\Delta N}$, respectively. the green dashed lines represent the hydrogen-bond between T303 and W307. EM densities for the SFs are presented in white surface contoured at 6 σ.

**f** The superposition of the SF of Na$_V$Eh$^{WT}$ (gray) and Na$_V$Eh$^{\Delta N}$ (red). The red and green arrows indicate the shifts of E305 and W307, respectively. **g** The top view of the fourfold symmetric SF of Na$_V$Ab$^{I217C}$ (PDB code: 3RVY). The black dashed square represents the SF size of Na$_V$Ab$^{I217C}$. **h** The top view of the two-fold symmetric SF of Na$_V$Ab$^{WT}$ (PDB code: 4EKW). The opposing two subunits are colored in white and green, respectively. The black and red dashed lines represent the SF size of Na$_V$Ab$^{WT}$. **i** The superposition of the SFs of Na$_V$Ab$^{I217C}$ (light purple) and Na$_V$Ab$^{WT}$ (gray). E177 residues are shown side-chain in sticks.

the VSD and employing meta-dynamics will provide valuable insights into the role of the VSD in the conformational changes between the active and inactivation states of Na$_V$ channels.

The EM structure of Na$_V$Eh$^{\Delta N}$ exhibits an expanded SF and a constricted activation gate. Complemented with the functional and the MD simulation results, we hypothesize that both the activation gate and the pore-loop region determine the slow inactivation of Na$_V$ channels. In the presence of the N-helix, the channel activates in response to depolarizing pulses, then the N-helix quickly blocks the open activation gate, resulting in the fast inactivation of Na$_V$Eh (Supplementary Fig. 12a–c). In the absence of the N-helix, the unstable open activation gate of Na$_V$Eh$^{\Delta N}$ could transit to a closed state that possibly contributes to the dramatically reduced peak current. The prolonged depolarizing stimuli eventually induce the conformational changes in the pore-loop region and drive the channel into the slow-inactivated state (Supplementary Fig. 12d).

## Discussion

Voltage-gated sodium (Na$_V$) channels govern the excitability of excitable cells. While the fast inactivation of eukaryotic Na$_V$ channels takes place in a time scale of several milliseconds in response to a single depolarizing pulse; the slow inactivation gradually reduces the number of Na$_V$ channels available for activation after receiving prolonged or repetitive stimuli. Therefore, slow inactivation regulates cellular excitability and is involved in a variety of physiological activities[9–11]. Although the molecular determinants and the underlying mechanisms for fast inactivation have been well-documented[12,13,16], the inscrutable process of slow inactivation remains poorly understood. In this study, we show that the fast inactivation particle (N-helix) removed Na$_V$Eh$^{\Delta N}$ exhibited the properties of slow inactivation with a small portion of fast inactivation component, positive shift in voltage-dependence of inactivation, and incomplete inactivation (Fig. 1), which resemble the behavior of human Na$_V$1.5 with a single mutation of F1485Q in the IFM-motif[35]. The residual fast inactivation component and the incomplete inactivation are most likely caused by the residual fast inactivation particles in Na$_V$Eh$^{\Delta N}$ and Na$_V$1.5 F1485Q mutant[12,35]. On the other hand, the homo-tetrameric prokaryotic Na$_V$ channels only show slow inactivation because of lacking the fast inactivation particles (IFM-motif or N-helix), many of which, however, possess a C-terminal helix forming a four-helix bundle[44,55–57]. The C-terminal helical bundle was found to

regulate the slow inactivation of prokaryotic Na$_V$ channels[55,58]. It would be interesting to know whether modifying prokaryotic Na$_V$ channels by adding the N-terminal loop of Na$_V$Eh and deleting the C-terminal helix could confer fast inactivation on prokaryotic Na$_V$ channels. These analyses suggest that Na$_V$Eh might be an evolutionary intermediate between the prokaryotic and eukaryotic Na$_V$ channels, narrowing their structural and functional gaps.

We further described the cryo-EM structure of Na$_V$Eh$^{\Delta N}$ in the potential slow-inactivated state. The structure demonstrated that the fourfold symmetric channel possesses activated VSDs, a dilated SF, and a closed activation gate, revealing a possible mechanism of Na$_V$ channel slow inactivation that is distinct from the symmetry collapse mechanism of Na$_V$Ab[29]. Although the activated VSDs of Na$_V$Eh$^{\Delta N}$ appeared to be identical to that of Na$_V$Eh$^{WT}$, the intracellular side of Na$_V$Eh$^{\Delta N}$ underwent conformational contraction of ~8 Å, resulting in the almost fully closed activation gate. Additionally, the EM densities clearly revealed that the P2 helix in the pore region underwent substantial conformational shifts, leading to the dilated SF. Further structural analysis indicated that the expanded SF is unfavorable for coordinating and conducting Na$^+$, strongly suggesting that the pore region could be a gate for slow inactivation. In fact, previous mutagenesis studies have shown that the SF of Na$_V$ channels is involved in Na$_V$ channel slow inactivation. For example, the mutation W402C in the SF of rat Na$_V$1.4 Domain I (corresponding to W307 in the SF of Na$_V$Eh) markedly attenuated slow inactivation[48]. The mutation V754I in DII-SF of Na$_V$1.4 (corresponding to L298 in the SF of Na$_V$Eh) impaired slow inactivation, while the mutation I891V at the corresponding position of Na$_V$1.5 enhanced slow inactivation[50]. Furthermore, conformational changes in the SFs of potassium channels were suggested to mediate the C-type slow inactivation[45,46]. The comparison of the SFs of the potassium channels and Na$_V$Eh$^{\Delta N}$ demonstrates that the distorted SFs are unfavorable for ion coordination and conductance, suggesting that the slow inactivation of Na$_V$ and potassium channels are functionally and structurally conserved to some extent. Moreover, previous studies have shown that the removal of the fast inactivation gate enhances the slow inactivation[36–38], presumably because the fast inactivation particles stabilize the activation gate[14,16]. In agreement with a large body of functional studies suggesting that the VSD, the pore region, the activation gate, and the fast inactivation gate are involved in the slow inactivation of Na$_V$ channel, our findings highlight that the slow inactivation of Na$_V$ channel is a complicated process that may result from the mutual interactions of the VSD, the SF, the activation gate, and the fast inactivation gate.

## Methods
### Whole-cell voltage-clamp recordings
HEK293T (ATCC, CRL-3216) cells were maintained in Dulbecco's Modified Eagle Medium (DMEM, Gibco, USA) supplemented with 10%(v/v) Fetal Bovine Serum (FBS, PAN-Biotech, Germany) at 37°C with 5% CO$_2$. The cells were grown in the culture dishes ($d$ = 3.5 cm) (Thermo Fisher Scientific) for 24 h and then transfected with plasmids of Na$_V$Eh$^{WT}$ or mutants using lipofectamine 2000 (Thermo Fisher Scientific, USA). The P2 viruses of Na$_V$Eh$^{\Delta N}$ were obtained from Sf 9 (Invitrogen, USA) insect cells and were used to transfect HEK293T cells for protein expression[59]. 24–48 h after transfection, whole-cell voltage-clamp recordings were obtained using a HEKA EPC-10 patch clamp amplifier (HEKA Electronic, Germany) and PatchMaster software (HEKA Electronic, Germany). Extracellular and intracellular (pipette) solutions are given in Supplementary Table 2. The pipettes were fabricated by a DMZ Universal Electrode puller (Zeitz Instruments, Germany) using borosilicate glass to resistances of 1.5–2.5 MΩ. Whole-cell voltage-clamp recordings were made from isolated, GFP-positive cells at room temperature. The currents were acquired at a 50 kHz sample rate, and series resistance ($R_s$) compensation was set to 70–90%.

To characterize the inactivation properties of Na$_V$Eh$^{\Delta N}$ channels, cells were held at −150 mV, and then a series of 30 s voltage steps from −120 mV to 0 mV in 10 mV increments followed by 100 ms test pulse at −20 mV were applied. Before the 30 s pre-pulse, 100 ms reference-pulse at −20 mV were applied, followed by a return to −150 mV for 1 s to recover from fast inactivation. To characterize the activation properties, cells were held at −150 mV and currents were elicited by 100 ms depolarizing pulses between −120 mV and 0 mV in steps of 10 mV. To assess the recovery from slow inactivation, we initiated currents through a pre-pulse at −20 mV for 30 s, followed by an inter-pulse that varied from 1 ms to 33 s at −150 mV. Subsequently, a 100 ms test pulse at −20 mV after the inter-pulse was applied.

All data reported as mean ± SEM. Data analyses were performed using Origin 2020 (OriginLab, USA), Excel 2019 (Microsoft, USA), and GraphPad Prism 8.0.2 (GraphPad Software, USA).

Steady-state activation ($I$–$V$) curves were generated using a Boltzmann equation:

$$\frac{G}{G_{max}} = \frac{1}{1+ \exp\left[(V - V_{0.5})/k\right]} \qquad (1)$$

where $G$ is the conductance, $G_{max}$ is the maximal slope conductance, $V$ is the test potential, $V_{0.5}$ is the half-maximal activation potential, and $k$ is the slope factor.

Steady-state inactivation curves were generated using a Boltzmann equation:

$$\frac{I}{I_{max}} = \frac{1}{1+ \exp[(V - V_{0.5})/k]} \qquad (2)$$

where $I$ is the current at indicated test pulse, $I_{max}$ is the maximal current during the reference-pulse, $V$ is the test pulse, $V_{0.5}$ is the half-maximal inactivation potential and $k$ is the slope factor.

Recovery curves from slow inactivation were fit using a double exponential of the following equation:

$$\frac{I_{test}}{I_{pre}} = A_1 * \exp\left(-\frac{t - K}{\tau_1}\right) + A_2 * \exp\left(-\frac{t - K}{\tau_2}\right) + C \qquad (3)$$

where $A_1$ and $A_2$ represent the amplitudes at the start of the fit region of $\tau_1$ and $\tau_2$, which are the time constants for inactivation, $K$ is the time shift, and $C$ is the steady-state asymptote.

### Expression and purification of Na$_V$Eh
The purification of Na$_V$Eh$^{WT}$ and Na$_V$Eh$^{\Delta N}$ were prepared similarly to our previous study[16]. In brief, baculoviruses were generated by using the standard bac-to-bac system (Invitrogen) in Sf 9 insect cells. HEK293F (Gibco, FreeStyle 293-F) cells were used to produce Na$_V$Eh proteins. HEK293F cells were cultured in OPM-293 medium (OPM) and were transfected by P2 viruses at a ratio of 1:100 (v/v) when cell density reached $2.5 \times 10^6$ per mL. The cells were cultured for another 48 hours before harvesting. For the purification of Na$_V$Eh$^{\Delta N}$, cell membrane was collected by ultracentrifugation and was solubilized in Buffer A (20 mM HEPES pH 7.5, 150 mM NaCl, 2 mM β-mercaptoethanol (β-ME), as well as a protease inhibitor cocktail including 1 mM phenylmethylsulfonic acid Acyl fluoride (PMSF), 0.8 μM pepstatin, 2 μM leupeptin, 2 μM aprotinin, and 1 mM benzamidine) supplemented with 1% (w/v) n-dodecyl-β-ᴅ-maltoside (DDM), 0.15% (w/v) cholesterol Hemisuccinate (CHS), 5 mM MgCl$_2$, and 1 mM ATP. The supernatant of solubilization was incubated with Strep-Tactin beads (Smart-Lifesciences, China) pre-equilibrated with buffer B (buffer A supplemented with 5 mM MgCl$_2$, 5 mM ATP and 0.06% (w/v) Glyco-diosgenin (GDN) (Anatrace, USA)). The beads were washed, and the protein was eluted with 5 mL buffer C (buffer B plus 5 mM desthiobiotin (Sigma, USA)). The eluted samples were further purified by running through a

Superose Increase 10/300 GL (GE healthcare, USA) column pre-equilibrated with 20 mM HEPES, 150 mM NaCl, 0.007% GDN (w/v), and 2 mM β-mercaptoethanol (β-ME), pH 7.5. Peak fractions were collected and concentrated to 7.8 mg/mL. For the purification of $Na_VEh^{WT-Ca}$ and $Na_VEh^{WT-EGTA}$, all buffers were supplemented with 2 mM $Ca^{2+}$ and 2 mM EGTA, respectively.

## Cryo-EM sample preparation and data collection
Aliquots of 3.0 μL purified $Na_VEh$ samples were placed on glow-discharged holey copper grids (Quantifoil, Cu R1.2/1.3, 300-mesh). Then the grids were blotted for 2.5–3.5 s by filter papers and were plunge-frozen in liquid ethane cooled by liquid nitrogen using a Vitrobot Mark IV (Thermo Scientific, USA) with 100% humidity at 4 °C. All cryo-EM data were acquired using a Titan Krios transmission electron microscope (Thermo Scientific, USA) operated at 300 kV, equipped with a K2 Summit direct detector (Gatan, USA) and Quantum GIF energy filter (Gatan, USA) with a slit width of 20 eV. All movie stacks were collected using SerialEM with a physical pixel size of 1.04 Å (super-resolution mode). The defocus range was set between −1.2 and −2.2 μm. The dose rate was set at 10 counts/pixel/s. Each movie stack was exposed for 6.4 s and was fractionated into 32 frames with a total dose of $60e^-/Å^2$. A total of 2050, 1300, and 2237 movie stacks were collected for $Na_VEh^{WT-Ca}$, $Na_VEh^{WT-EGTA}$, and $Na_VEh^{ΔN}$, respectively.

## Data processing
All the movie stacks were motion-corrected, binned by 2-fold, and dose-weighted using MotionCorr2[60]. Defocus values of each summed micrograph were estimated by Gctf[61]. A total of 447,915, 352,310, and 454,312 particles were picked for $Na_VEh^{WT-Ca}$, $Na_VEh^{WT-EGTA}$, and $Na_VEh^{ΔN}$, respectively. All 2D classification, 3D classification, and particle polishing were carried out in Relion3[62]. The final data set of 110,665, 59,377, and 81,844 particles for $Na_VEh^{WT-Ca}$, $Na_VEh^{WT-EGTA}$, and $Na_VEh^{ΔN}$ were refined in CryoSPARC[63] to resolutions of 2.6 Å, 3.3 Å, and 3.1 Å, respectively. The detailed data processing flowcharts were presented in Supplementary Figs. 2 and 5.

## Model building
The $Na_VEh$ structure (PDB code: 7X5V) was used as an initial model and was manually fitted into the EM maps of $Na_VEh^{WT-Ca}$, $Na_VEh^{WT-EGTA}$, and $Na_VEh^{ΔN}$ using Chimera[64]. After being manually checked and corrected in COOT[65], the resulting models were refined in Phenix[66], respectively. The FSC curves of the refined models were calculated by Phenix.mtrage. The statistics of cryo-EM data collection and model refinement were summarized in Supplementary Table 1. All figures were prepared with PyMOL (Schrödinger, LLC), Prism 8.0.1 (GraphPad Software), and ChimeraX[67].

## Molecular dynamics simulations
The MD simulation setup followed the protocols outlined in our previous study[51], including the preparation of force fields for the protein, DMPC lipids, and ligands, as well as the control parameters. The simulated systems were solvated in a water environment containing 150 mM NaCl. The energy minimization procedure was initially conducted using the steepest descent method. Before production runs, six equilibrium steps were performed with a combined duration of 2 ns. All simulations were performed using the GROMACS 2021 suite of programs[68]. To capture the system's behavior, frames were saved at every nanosecond throughout the trajectories. The frames of each trajectory were clustered using GROMOS algorithm[69] by considering all the non-hydrogen atoms of the protein backbone with an RMSD cutoff of 1.5 Å. The diameters of the activation gate were calculated for each frame. Specifically, the initial model was oriented such that the axis of the removed N-helix is aligned with the Z axis using EDPDB[70]. Subsequently, the structures in each frame were superimposed onto the initial model to align their channels along the Z axis. The channel diameter was determined using HOLE[71]. To assess the open or closed state of the channel, the minimum diameter in the region extending from the channel center to the side involved in N-helix interaction was recorded as an indicator. Source data for details of system setup, simulation parameters, and trajectories are provided as a Source Data file.

## Reporting summary
Further information on research design is available in the Nature Portfolio Reporting Summary linked to this article.

## Data availability
The data that support the findings of this study are available from the corresponding author upon request. The cryo-EM maps have been deposited in the Electron Microscopy Data Bank (EMDB) under accession codes EMD-36042 ($Na_VEh^{ΔN}$); EMD-36039 ($Na_VEh^{WT-Ca}$); EMD-36041 ($Na_VEh^{WT-EGTA}$). The atomic coordinates have been deposited in the Protein Data Bank (PDB) under accession codes 8J7M ($Na_VEh^{ΔN}$); 8J7F ($Na_VEh^{WT-Ca}$); 8J7H ($Na_VEh^{WT-EGTA}$). $Na_VEh$ (MMETSP transcriptomic datasets [https://www.bco-dmo.org/dataset/665427] ID: CAM-PEP_0187654740, MMETSP0994-7). The PDB accession codes for the published structures used in this study are 3RVY (pre-open $Na_VAb$), 4EKW (inactivated $Na_VAb$), 6P6W (resting $Na_VAb$), 5HVX ($Na_VMs$), 6UZ0 (rat Nav1.5), 1K4C (KcsA), 3F5W (C-type inactivated KcsA), 7SIP (Shaker $K_V$ channel), 7SJ1 (Shaker $K_V^{W434F}$ channel), 7X5V (wild-type $Na_VEh$). The source data underlying Fig. 1e–g, Supplementary Figs. 1b and 4 are provided as a Source Data file. Source data are provided with this paper.

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

## Acknowledgements

We thank X. Huang, B. Zhu, X. Li, L. Chen, and other staff members at the Center for Biological Imaging (CBI), Core Facilities for Protein Science at the Institute of Biophysics, Chinese Academy of Science (IBP, CAS) for the support in cryo-EM data collection. We thank Wei Fan for her research assistant service. This work is funded by the National Natural Science Foundation of China (grant nos. 32271272 and T2221001 to D.J., U23A20143 and 32070031 to J.J., 82271498 to Z.H., 82071851 to J.G.), Science and Technology Innovation 2030 Major Project (grant no. 2021ZD0202103 to Z.H.), and the program for HUST Academic Frontier Youth Team (grant no. 5001170068 to J.G.) and Ningxia Hui Autonomous Region Key Research and Development Project (grant no. 2022BEG02042 to Z.H.).

## Author contributions

D.J. conceived and designed the experiments. H.C., J.Z. and R.Y. prepared the sample for the cryo-EM study and made all the constructs. H.C. and J.Z. collected cryo-EM data, processed the data, and built and refined the models. H.C. and J.D. prepared all figures. J.D. and Y.S. collected the electrophysiology data. B.H. and F.Z. performed the MD simulation analysis. J.J., J.G., Z.H. and D.J. analyzed and interpreted the results. Z.X. and D.J. wrote the paper, and all authors reviewed and revised the paper.

## Competing interests

The authors declare no competing interests.
