## [Peer Review File · Nature Communications]

Structural mechanism of voltage-gated sodium channel slow inactivationReviewer #1 (Remarks to the Author):

The manuscript describes interesting work on the determination of a cryo structure of NavEh sodium channel, potentially captured in the slow-inactivated state after the removal of the N-terminal helix. Electrophysiology experiments are performed to assess the functional properties of the DeltaN channel.

The quality of the determined structure allows for conjectures on its functional state, mostly based on comparison with previously published conformations. MD simulations are performed on the WT structure by removing the N-terminal helix.

The authors thus propose that the determined structure is representative of the channel slow inactivated state.

I find the work well executed and the paper well written in general. I would in principle support publication as I agree the structure is potentially interesting, however I have a major concern about its functional annotation.

It appears to me that an evidence that the resulting structure is in the inactive state is still lacking. How can the authors be sure that it is not a non-physiological structure? Can mutations be designed based on the determined structure to clarify its functional state?

Also, they perform MD simulations of the WT channel structure after removing the pore-blocking N-terminal helix, and observe collapse of the activation gate. But what does this prove about the functional state of the cryo-DeltaN? The observation that the simulated DeltaN-WT structure does not reach the cryo-DeltaN conformation at the SF is not enough to confirm the state of the latter.

I think they should simulate their cryo structure, and see how it behaves on the timescale studied for the WT, in particular concerning the SF conformations.

Reviewer #2 (Remarks to the Author):

Chen H. et al. showed in a very interesting and nice work that the removal of the N-helix of NaVEh (NaVEh Δ N) results in significant conformational changes of the channel which they proposed to be a slow inactivated state of NavEh channel. They presented a cryo-EM structure of the NaVEh Δ N in a plausible slow-inactivated state, where they featured a closed activation gate and a dilated selectivity filter (SF). They also conducted e-phys experiments to confirm that the channel lacks its fast inactivation ability, but still retains slow inactivation. Molecular dynamics simulations suggest that the open inner gate is inherently unstable and quickly transits to the closed state. Overall, their results provide important and interesting mechanistic insights into NaV channel slow inactivation.

Concerns:

- Despite that the structure of NaVEh Δ N attributes a convincing plausible slow inactivated state with a wide selectivity filter and a closed activation gate. The functional analysis does not support that. For example, in mammalian VGSC, removing the fast inactivation particle (IFM) facilitates slow inactivation to occur with a steeper voltage dependence with a higher degree of completeness (Please check Featherstone et al., 1996, and Richmond et al., 1998). I do not see that in NaVEh Δ N. Instead, the authors reported 30% inactivation-resistant channels, with a rightward shift in the steady-state inactivation profile, indicating a strong resistance to inactivation.
- The raw traces of NaVEh Δ N show a clear component of fast inactivation at the very few milliseconds of the depolarizing pulses followed by a flat current, which shows that there is still a fast inactivation component. I wonder if you cut more residues of the N-terminus, would that lead to more ablation of the fast inactivation machinery?
- NaVEh Δ N inactivation machinery appears to be an evolutionary intermediate between the

prokaryotic and the mammalian Nav, the authors should elaborate more on this evolution transition. There are many papers that discuss the molecular mechanisms of bacterial and mammalian channels that should be used, e.g., Irie et al, JBC, 2011, Irie K. et al., Nat. comm.,2012, Gamal El-Din et al, 2013, 2015, JGP, and many other papers.

- References need careful revision by authors. The authors for example talked about fenestrations and its changes during gating and they did not cite important papers like Gamal El-Din et al., 2019, PNAS. Ghovanloo, M. R., 2020, eLife.

Reviewer #3 (Remarks to the Author):

In this report by Chen, Xia, Dong, Huang and colleagues, the authors build upon previous work, presenting cryo-EM structures of a voltage-gated sodium channel from coccolithophore *Emiliana huxleyi* along with complementary electrophysiological and computational analyses. In their previous study, the N-termini of NavEh was found to be responsible for the fast activation of NavEh. By deleting the N-termini, the authors find here that the helical densities that occlude the ion permeation pathway in the full-length channel are no longer present. However, the channel does not adopt an open state, instead the pore-lining S6 helices collapse inward to seal the activation gate. Additional conformational changes are also found in the selectivity filter. Electrophysiological analyses reveal that the N-terminal deletion mutation is active and does not display fast inactivation. However, it's activity is much reduced compared to wild-type NavEh. Based on these electrophysiological analyses, the authors assign the structure of the N-terminal deletion mutation as a slow inactivated state. However, I have concerns regarding the assignment of the state and the functional relevance of this slow inactivated state. For these reasons, which are detailed below, the findings in the manuscript are insufficient to support the claim that the authors "results provide important mechanistic insights into NaV channel slow-inactivation" and I am therefore unwilling to support its publication in Nature Communications at this time.

Major concerns:

1. The section is "Ca²⁺ independent activation of NaVEh" is completely out of place and should be removed the manuscript. It does not contribute to the understanding of slow inactivation, which is the focus of this manuscript.
2. It is not clear from the traces presented in Figure 2 that N-terminal deletion of NaVEh undergoes slow inactivation. The authors should quantify the kinetics of inactivation as a function of voltage to compare with other voltage-gated sodium channels that are known to undergo slow inactivation to demonstrate that the phenomenon described here corresponds to slow inactivation. Such a demonstration is imperative as the inactivation described here is "composed of two components, a relatively fast component within the first 10 ms and a following prolonged slow reduction of the current"
3. Interestingly, the inactivation described here does not appear to be complete. Longer pulses should be performed to determine if inactivation can reach completion. Otherwise, there would be a disconnect between the structural analysis, which only revealed closed channels, and the electrophysiological analyses, in which about 40% of the channels were still open at steady state.
4. Similarly, longer pulses should be performed to more completely characterize the recovery from inactivation.
5. Is the structure of NaVEhWT is taken from a previous manuscript? If so, that should be explicitly defined in the text.
6. The authors suggest the structure of NaVEhdN corresponds to a slow inactivated state. However, the electrophysiological data suggests that slow inactivation is incomplete. Are there alternative low-abundance states that were discarded during classification? Additionally, the described slow inactivation process is composed of two components. To which of these phases does the resolved structure correspond and how was such an assignment made?

7. The selectivity filter of NaVEhdN adopts a distorted state, which based on comparisons with potassium channels, leads the authors to propose that the selectivity filter contributes to the inactivated state. The changes in the selectivity filter are subtle. At the moderate resolution of the structure, it would be helpful to include figures in the main text showing the raw density for the selectivity filter to aid the readers in understanding the changes. Especially when calling into significance a movement of 0.2 Å in each monomer for the backbone oxygen of G304.

8. Continuing from point 7, it is unclear that the distortions described in the selectivity filter would prevent ion permeation. A much more thorough evaluation of the effect of these changes on ion permeation should be performed as the selectivity filters of potassium channels are quite distinct from those of sodium channels and it is not clear that the distortions would lead to similar functional effects.

9. The timescales for the closure of the activation gate are much faster than would be expected for slow inactivation. In the replicates presented in Figure 6, all of the activation gates collapse within 700 ns, whereas currents can be measured from channels in cells after 10 seconds. The collapse in the simulations thus does not appear to correspond to the slow inactivation seen in electrophysiological recordings.

10. Altogether, the functional state to which the determined structure corresponds is not clear from the analyses provided. It is essential that the authors uncover the functional state of this structure and elucidate the roles of the selectivity filter and activation gate in slow inactivation.

Minor Concerns:

1. Figure 1d – Y282 appears to be a threonine
2. What is the red dashed box in Figure 3d-e?

Response to Reviewers' Comments

We sincerely thank the referees for their time to evaluate our manuscript, as well as for their valuable suggestions and insightful criticisms that have helped to improve the manuscript. In accordance to the referees' suggestions, we have made the following revisions:

1. To focus on slow inactivation, we have moved the old Figure 1 into supplementary information as the new Supplementary Figure 1.
2. We have performed additional electrophysiological experiments to show that the residual fast inactivation component is mediated by the residual N-terminal fast inactivation particle. We have also improved the New Figure 1 to better elucidate the functional properties of NavEh^{ΔN}.
3. We have performed MD simulations using the pore domain of NavEh^{ΔN}, which showed similar results to NavEh^{WT} due to the limitations of current MD systems. Thus, we have removed the MD section in the revised our manuscript.
4. We have also carefully revised the manuscript by adding necessary references, comparison with human Nav channels, and discussion.

A detailed point-by-point response in black to the referees' comments in blue is provided below.

Reviewers' Comments:

Reviewer #1 (Remarks to the Author):

The manuscript describes interesting work on the determination of a cryo structure of NavEh sodium channel, potentially captured in the slow-inactivated state after the removal of the N-terminal helix. Electrophysiology experiments are performed to assess the functional properties of the DeltaN channel. The quality of the determined structure allows for conjectures on its functional state, mostly based on comparison with previously published conformations. MD simulations are performed on the WT structure by removing the N-terminal helix. The authors thus propose that the determined structure is representative of the channel slow inactivated state. I find the work well executed and the paper well written in general. I would in principle support publication as I agree the structure is potentially interesting, however I have a major concern about its functional annotation. It appears to me that an evidence that the resulting structure is in the inactive state is still lacking.

Reply: We thank Reviewer 1's positive comments on the quality of our work, and for his/her suggestions to improve our manuscript.

How can the authors be sure that it is not a non-physiological structure?

Reply: We thank Reviewer 1 for raising this point, which was also concerned by Reviewer 3. We proposed the cryo-EM structure was captured in a potential inactivated-state for the following reasons:

1. The voltage-sensing domain (VSD) of NavEh^{ΔN} exhibited activated conformation, which is nearly identical to that of our previous NavEh^{WT} and very similar to the activated VSDs

of other Nav channels. Meanwhile, the SF and the activation gate of NavEh^{ΔN} are significantly different from that of NavEh^{WT}. The observed activated VSD and non-conductive pore of NavEh^{ΔN} coincide with the characteristics of slow inactivation (See **Response Figure 1a-f** below).

2. E-phys results showed that NavEh^{ΔN} exhibits slow inactivation properties which is sharply distinct from the fast inactivation of NavEh^{WT}. (**Response Figure 1g-h** and **new Figure 1**). The residual fast inactivation component, positive shift in voltage-dependence of inactivation and incomplete inactivation of NavEh^{ΔN} resemble slow inactivation of human Nav1.5 with the F1485Q mutation in the IFM-motif (PMID: 8013069).

3. The purification of NavEh^{ΔN} were performed at 0 mV for over 10 hours, which would facilitate the NavEh^{ΔN} protein assuming a slow-inactivated conformation.

We also agree with Reviewer 1 that the slow-inactivation of Nav channels is a complicated process and its state(s) is difficult to firmly identify. The structure of NavEh^{ΔN}, which may represent a possible slow-inactivated state of NavEh, provides mechanistic insights into the slow inactivation of Nav channels. We have toned down the statement of “NavEh^{ΔN} in a slow-inactivated state” as “NavEh^{ΔN} in a potential slow-inactivated state” in the revised manuscript.

Response Figure 1: a-c, VSD comparison of NavEh^{ΔN} with NavEh^{WT} (a), bacterial NavAb (b), and rat Nav1.5 (c). d-f, Activation gates of NavEh^{ΔN} (d), bacterial NavMs (e), and rat Nav1.5 (f). g-h, Example current traces of NavEh^{WT} (g) and NavEh^{ΔN} (h).

Can mutations be designed based on the determined structure to clarify its functional state?

Reply: We thank Reviewer 1's suggestion on this. We have performed mutagenesis studies to identify the functional state of this construct. To clarify the residual component of fast inactivation of NavEh^{ΔN}, we generated a series of mutants with varied N-terminal deletions (**Response Figure 2a**). Although those mutants expressed similarly to NavEh^{Δ13} in HEK293 cells, no current could be observed (**Response Figure 2e-g**). We further mutated ¹⁵AAAA¹⁸ of NavEh^{Δ13} to ¹⁵EEEE¹⁸, which would increase the repulsion of the N-terminal loop with the negatively charged outer mouth of the activation gate. The resulting

construct NavEh^{ΔN18E} exhibited almost completely ablation of fast inactivation (**Response Figure 2d**), in consistence with the IFM-QQQ mutant of human Nav channel. Those results showed that the residual fast inactivation is indeed caused by the residual N-terminal fast inactivation particle (**Response Figure 2a-g**). To verify the effects of the conformational changes in P2 helix, we generated the mutant P314A; however, this construct turned out to be non-functional (**Response Figure 2i**), suggesting the importance of this residue and the P2 helix for NavEh function. In addition, a previous study showed that Nav1.5 exhibits more resistance to slow inactivation than Nav1.4, the substitution of V754 (corresponding to L298 in the SF of NavEh) in Nav1.4 DII-SF region with an Ile at the corresponding position of Nav1.5 enhanced the resistance to slow inactivation of Nav1.4 similar to Nav1.5 (PMID: 11325725). We therefore generated a NavEh^{ΔN} variant with a L298V mutation, which also showed no current (**Response Figure 2h**), presumably because the homotetrameric organization of NavEh carries four L298V mutations in the pore domain, rather than only one mutation in the asymmetric human Nav channels. Furthermore, we have improved the E-phys results of NavEh^{ΔN} (**new Figure 1**) by increasing the expression levels and extending condition pulse to better elucidate the slow inactivation properties of NavEh^{ΔN} in the revised manuscript.

Response Figure 2: The mutants with varied N-terminal truncations and the example current traces of NavEh^{WT} (b), NavEh^{ΔN13} (c), NavEh^{ΔN18E} (d), NavEh^{ΔN21} (e), NavEh^{ΔN48} (f), NavEh^{ΔN59} (g), and NavEh^{L298V} (h), NavEh^{P314A} (i).

Also, they perform MD simulations of the WT channel structure after removing the pore-blocking N-terminal helix, and observe collapse of the activation gate. But what does this prove about the functional state of the cryo-DeltaN?

Reply: We thank Reviewer 1's comments and suggestions on MD results. While we did perform MD studies on NavEh^{ΔN} as suggested by the reviewer, we realized that the

duration of our simulations, limited to the microsecond timescale, may not be sufficient to fully capture the desired conformational changes between the open and slow-inactivated states, which may occur over a longer timescale of several seconds. Consequently, the results obtained from these MD studies may not sufficiently explain the observations from the experiments. In light of these considerations, we have made the decision to exclude the MD-related sections from the revised manuscript.

The rationale behind this decision together with our next-step plan are detailed as follows:

Initially, we aimed to use MD simulations to investigate the conformational changes between the open state (represented by NavEh^{WT} structure with N-helix removed) and the slow-inactivated state (represented by NavEh^{ΔN} structure). Six independent MD trajectories, each lasting 3-4 microseconds, were performed on the pore domains of the NavEh^{WT} and NavEh^{ΔN} structures.

After analysis of the simulations, we did observe closed activation gate from both simulations which are consistent with previous E-phys results and MD simulation results of mammalian Nav1.5 (PMID: 34520724; 7876824); but unfortunately, we did not observe any significant conformational changes in the P2 helix within the selectivity filter (**Response Figure 3**). Additionally, the observed changes in the channel radii of active state could not explain the experimental observations (Figure 6a in the previous version of manuscript).

Response Figure 3. P2 helix conformation dynamics during MD study. The distance between Ca atom of A310 and Ca atom of P314 is used to indicate the conformation of P2 helix. When P2 helix adopts a standard alpha helix conformation in NavEh^{WT}, the Ca distance between A310 and P314 is around 7 Å. When P2 helix adopts a helix-loop-helix conformation in NavEh^{ΔN}, the Ca distance between A310 and P314 is around 11 Å.

We speculate that two factors may have contributed to the inability of our MD study to establish a connection between the open and slow-inactivated states. The first factor is the limited timescale of our sampling strategy, which was constrained to microsecond timescales. In contrast, based on experimental findings, the conformational changes between the states may require a timescale of seconds from the electrophysiological experiments. To address this limitation, we are actively exploring enhanced sampling

methods, such as meta-dynamics. The second factor is the truncation of the voltage-sensing domain (VSD) during our MD simulations. It is possible that this truncation impacted the integrity of the whole system and thereby block the desired conformational changes. However, including the four VSDs in our current sampling process would significantly increase the computational resources required. Nevertheless, we anticipate that future MD simulations incorporating the VSD and employing meta-dynamics might provide valuable insights into the dynamics in the conformational changes between the open and slow-inactivated states, as well as the role of the VSD in this process.

Considering these limitations, we have opted to remove the MD contents from the current manuscript. This decision will allow us to dedicate more time and employ new sampling methods to conduct further MD studies on the revised systems including the VSDs. We aim to publish these results in future studies, which will further enhance our understanding of the system.

The observation that the simulated DeltaN-WT structure does not reach the cryo-DeltaN conformation at the SF is not enough to confirm the state of the latter. I think they should simulate their cryo structure, and see how it behaves on the timescale studied for the WT, in particular concerning the SF conformations.

Reply: We thank Reviewer 1's suggestion on MD simulations. We have performed similar MD simulations using the pore domain of NavEh^{ΔN} as an initial model. As we have clarified above, due to the limitations of timescales and lacking the VSDs in the current MD systems, we did not observe the dynamics in the P2 helix region. We have decided to remove the MD section from the revised manuscript. In the future, we will optimize the MD systems to study the dynamics of the key states in Nav channel activation and inactivation.

Reviewer #2 (Remarks to the Author):

Chen H. et al. showed in a very interesting and nice work that the removal of the N-helix of NavEh (NavEh^{ΔN}) results in significant conformational changes of the channel which they proposed to be a slow inactivated state of NavEh channel. They presented a cryo-EM structure of the NavEh^{ΔN} in a plausible slow-inactivated state, where they featured a closed activation gate and a dilated selectivity filter (SF). They also conducted e-phys experiments to confirm that the channel lacks its fast inactivation ability, but still retains slow inactivation. Molecular dynamics simulations suggest that the open inner gate is inherently unstable and quickly transits to the closed state. Overall, their results provide important and interesting mechanistic insights into Nav channel slow inactivation.

Reply: We appreciate Reviewer 2's positive comments on the significance and the quality of our work, and for his/her suggestions to improve our manuscript.

Concerns:

- Despite that the structure of NavEh^{ΔN} attributes a convincing plausible slow inactivated state with a wide selectivity filter and a closed activation gate. The functional analysis does

not support that. For example, in mammalian VGSC, removing the fast inactivation particle (IFM) facilitates slow inactivation to occur with a steeper voltage dependence with a higher degree of completeness (Please check Featherstone et al., 1996, and Richmond et al., 1998). I do not see that in NavEh Δ N. Instead, the authors reported 30% inactivation-resistant channels, with a rightward shift in the steady-state inactivation profile, indicating a strong resistance to inactivation.

Reply: We thank Reviewer 2 for pointing out this. The removal of the fast inactivation particle of mammalian Nav channels (IFM to QQQ) indeed significantly increases the component of slow-inactivation, in comparison to that of wild-type channel. One possible explanation is that the binding of IFM-motif to its receptor site adjacent to the activation gate or the binding of N-helix within the activation gate stabilize the activation gates; removing the fast inactivation particles might facilitate the conformational changes in the activation gate when involving in slow inactivation process. Interestingly, the resistance to slow-inactivation of IFM-QQQ and the F-Q mutants of cardiac Nav channel are different, ranging from 15% inactivation-resistance for the IFM-QQQ mutant (J.E. Richmond et al, PMID: 9635748) to 30% inactivation-resistance for the F-Q mutant (H.A. Heartmann et al, PMID: 8013069), presumably because of the incomplete removal of the F-Q particle from the receptor site. In line with this, the F-Q mutant retained a large degree of fast inactivation component, and exhibited non-changed voltage-dependent activation but largely right shifted (~25 mV) voltage-dependent inactivation, those properties are consistent with that of NavEh Δ N (New **Figure 1**). We have also shown that the residual fast inactivation component of NavEh Δ N is caused by the residual N-terminal fast inactivation particle (See **Response Figure 4** below). Therefore, we reason that the resistance to inactivation of NavEh Δ N is due to the residual N-helix which may still interact with the activation gate and affect inactivation.

• The raw traces of NavEh Δ N show a clear component of fast inactivation at the very few milliseconds of the depolarizing pulses followed by a flat current, which shows that there is still a fast inactivation component. I wonder if you cut more residues of the N-terminus, would that lead to more ablation of the fast inactivation machinery?

Reply: We thank Reviewer 2's suggestion on the fast inactivation component of NavEh Δ N¹³ (deletion of residues 2-13), which was also pointed out by Reviewer 3. In fact, the residual component of fast inactivation of NavEh Δ N¹³ resembles the effects of the F1489Q mutation in brain Nav channel (15% of residual fast inactivation: J.W. West et al, PMID: 1332060) and the F1485Q mutation in heart Nav channel (70% of residual fast inactivation: H.A. Heartmann et al, 1994, PMID: 8013069), while mutation of the IFM-motif to QQQ almost completely removed fast inactivation (PMID: 1332060). To verify the cause of the observed component of fast inactivation, we made mutants with longer N-terminal deletion of delta 2-21 (NavEh Δ 21), delta 2-48 (NavEh Δ 48), delta 2-59 (NavEh Δ 59). Although those mutants expressed similarly to NavEh Δ 13 in HEK293 cells, no current could be observed (**Response Figure 4**). We further mutated ¹⁵AAAA¹⁸ of NavEh Δ 13 to ¹⁵EEEE¹⁸, which would increase the repulsion of the N-terminal loop with the negatively charged outer mouth of the activation gate. The resulting construct NavEh Δ N^{18E} exhibited almost completely ablation of fast inactivation (**Response Figure 4d**), in consistence with the IFM-QQQ mutant of

human Nav channel. These results indicate that the residual fast inactivation of NavEh^{Δ13} is still mediated by the residual N-terminal fast inactivation particle.

Response Figure 4: The mutants with varied N-terminal truncations and the example current traces of NavEh^{WT} (b), NavEh^{ΔN13} (c), NavEh^{ΔN18E} (d), NavEh^{ΔN21} (e), NavEh^{ΔN48} (f), and NavEh^{ΔN59} (g).

- NavEh^{ΔN} inactivation machinery appears to be an evolutionary intermediate between the prokaryotic and the mammalian Nav, the authors should elaborate more on this evolution transition. There are many papers that discuss the molecular mechanisms of bacterial and mammalian channels that should be used, e.g., Irie et al, JBC, 2011, Irie K. et al., Nat. comm., 2012, Gamal El-Din et al, 2013, 2015, JGP, and many other papers.

Reply: We agree with Reviewer 2 on this point. We have added a discussion of the relationship of NavEh inactivation between prokaryotic and the mammalian Nav channels and cited those related references in the revised manuscript (Lines 271-286).

- References need careful revision by authors. The authors for example talked about fenestrations and its changes during gating and they did not cite important papers like Gamal El-Din et al., 2019, PNAS. Ghovanloo, M. R., 2020, eLife.

Reply: We thank Reviewer 2's suggestion on the references. The references (Gamal El-Din T.M. *et al*, PNAS, PMID: 30518562; Sait, L. G., Sula, A., Ghovanloo, M. R. *et al*, Elife, PMID: 33089780; Gamal El-Din T.M. *et al*, 2022, PMID: 35222049;) for the fenestrations have been added and other references have been carefully checked in the revised manuscript (Lines 177-181).

Reviewer #3 (Remarks to the Author):

In this report by Chen, Xia, Dong, Huang and colleagues, the authors build upon previous work, presenting cryo-EM structures of a voltage-gated sodium channel from coccolithophore *Emiliana huxleyi* along with complementary electrophysiological and computational analyses. In their previous study, the N-termini of NavEh was found to be responsible for the fast activation of NavEh. By deleting the N-termini, the authors find here that the helical densities that occlude the ion permeation pathway in the full-length channel are no longer present. However, the channel does not adopt an open state, instead the pore-lining S6 helices collapse inward to seal the activation gate. Additional conformational changes are also found in the selectivity filter. Electrophysiological analyses reveal that the N-terminal deletion mutation is active and does not display fast inactivation. However, its activity is much reduced compared to wild-type NavEh. Based on these electrophysiological analyses, the authors assign the structure of the N-terminal deletion mutation as a slow inactivated state. However, I have concerns regarding the assignment of the state and the functional relevance of this slow inactivated state. For these reasons, which are detailed below, the findings in the manuscript are insufficient to support the claim that the authors “results provide important mechanistic insights into NaV channel slow-inactivation” and I am therefore unwilling to support its publication in Nature Communications at this time.

Reply: We appreciate Reviewer 3’s comments on our work, and for his/her suggestions to improve our manuscript.

Major concerns:

1. The section is “Ca²⁺ independent activation of NavEh” is completely out of place and should be removed the manuscript. It does not contribute to the understanding of slow inactivation, which is the focus of this manuscript.

Reply: We agree with Reviewer 3 that the “Ca²⁺ independent activation of NavEh” section does not directly contribute to the understanding of the slow inactivation. However, a previous study suggested a possible Ca²⁺ dependent activation of NavEh (K.E. Helliwell, 2020, PMID: 33004614). Before this study, it was not clear whether Ca²⁺ affects the activation and/or inactivation of NavEh. To investigate the molecular mechanism of slow inactivation using NavEh, it is necessary to determine this possible Ca²⁺ regulation. To focus on the slow inactivation as suggested by Reviewer 3, we have shortened this section and integrated it into the “Functional characterization of NavEh^{ΔN}” section (Lines 97-108), and moved the **old Figure 1** into supplementary as the new **Supplementary Figure 1** in the revised manuscript.

2. It is not clear from the traces presented in Figure 2 that N-terminal deletion of NavEh undergoes slow inactivation. The authors should quantify the kinetics of inactivation as a function of voltage to compare with other voltage-gated sodium channels that are known to undergo slow inactivation to demonstrate that the phenomenon described here corresponds to slow inactivation. Such a demonstration is imperative as the inactivation described here is “composed of two components, a relatively fast component within the first 10 ms and a following prolonged slow reduction of the current”

Reply: We thank Reviewer 3's comments on this. The current traces from NavEh^{ΔN13} show slowly decreased current amplitude, which is sharply distinct from the fast inactivation of NavEh^{WT} (**Response Figure 5b-c**). The I/I_{max} curve also shows voltage-dependent inactivation of NavEh^{ΔN13} (the 30% resistance to inactivation is discussed in Response 3 below). Those results demonstrate that NavEh^{ΔN13} indeed undergoes slow inactivation. In mammalian Nav channels, the substitution of the fast inactivation particle IFM-motif with QQQ almost completely removed fast inactivation (PMID: 1332060); however, the mutants with a single-site mutation of F1489Q in brain Nav channel (PMID: 1332060) and F1485Q in heart Nav1.5 channel (PMID: 8013069) retained a component of fast inactivation up to 15% and 70%, respectively. To verify the cause of the observed component of fast inactivation in NavEh^{ΔN13}, we made mutants with longer N-terminal deletions of delta 2-21 (NavEh^{ΔN21}), delta 2-48 (NavEh^{ΔN48}), delta 2-59 (NavEh^{ΔN59}). Although those mutants expressed similarly to NavEh^{ΔN13} in HEK293 cells, no current could be observed (**Response Figure 5**). We further mutated ¹⁵AAAA¹⁸ of NavEh^{ΔN13} to ¹⁵EEEE¹⁸, which would increase the repulsion of the N-terminal loop with the negatively charged outer mouth of the activation gate (PMID: 35581266). The resulting construct NavEh^{ΔN18E} exhibited almost completely ablation of fast inactivation (**Response Figure 5d**), in consistence with the IFM-QQQ mutant of human Nav channel. These results indicate that, similar to the F1489Q and F1485Q mutants of mammalian Nav channels, the residual fast inactivation of NavEh^{ΔN13} is most likely caused by the weak binding of the residual fast inactivation particles to the corresponding receptor sites. In fact, the residual fast inactivation component, positive shift in voltage-dependence of inactivation, and 30% resistance to inactivation of NavEh^{ΔN} closely resemble the properties of the Nav1.5 F1485Q mutant, which was shown to undergo slow inactivation (PMID: 8013069). We have added those functional comparisons in the revised manuscript.

Response Figure 5: The mutants with varied N-terminal truncations and the example

current traces of NavEh^{WT} (b), NavEh^{ΔN13} (c), NavEh^{ΔN18E} (d), NavEh^{ΔN21} (e), NavEh^{ΔN48} (f), and NavEh^{ΔN59} (g).

3. Interestingly, the inactivation described here does not appear to be complete. Longer pulses should be performed to determine if inactivation can reach completion. Otherwise, there would be a disconnect between the structural analysis, which only revealed closed channels, and the electrophysiological analyses, in which about 40% of the channels were still open at steady state.

Reply: We agree with Reviewer 3 that the inactivation of NavEh^{ΔN13} is incomplete. Even when increasing the conditioning pre-pulse to 30 s as suggested, there is still ~30% current resistant to inactivation. A similar resistance to slow inactivation was also observed in mammalian Nav channels with IFM-QQQ or F-Q mutations. The IFM-QQQ mutant showed 15% resistance to inactivation (J.E. Richmond et al, PMID: 9635748) and the F1485Q mutant displayed a large degree of 30% resistance to inactivation (H.A. Heartmann et al, PMID: 8013069). One possible explanation is that the binding of IFM-motif to its receptor site adjacent to the activation gate or the binding of N-helix within the activation gate stabilize the activation gates; deleting the fast inactivation particles might facilitate the conformational changes in the activation gate when involving in slow inactivation process. For NavEh^{ΔN13} and the F1485Q mutant of human Nav_v1.5 channel (PMID: 8013069), the residual component of fast inactivation suggested that the removal of the fast inactivation particle is incomplete, which may affect the activation gate and impair the slow-inactivation process. We have clarified this in the revised manuscript.

4. Similarly, longer pulses should be performed to more completely characterize the recovery from inactivation.

Reply: We thank Reviewer 3's suggestion on this. As suggested, we have performed electrophysiological recording experiments using the revised protocols with increased conditioning pre-pulse time of 30 s. The results of voltage-dependence of slow inactivation and the recovery from inactivation have been included in the New **Figure 1**. For Fig. 1a-b, we present three current traces (-40 mV, -20 mV, and 0 mV) to better elucidate the current differences between NavEh^{ΔN13} and NavEh^{WT}. The current density of NavEh^{ΔN13} were increased by ~40% when we used P2 viruses for cell transfection. However, the current density of NavEh^{ΔN13} is about one ninth of NavEh^{WT} (Fig. 1e), which is in line with the observation that the Nav_v1.5 F1485Q mutant exhibited one fifth current density of WT Nav_v1.5 (PMID: 8013069). For measuring steady-state voltage-dependence of slow inactivation as presented in Fig. 1c, we have increased the conditioning pre-pulse time to 30 s. The potential reason for this incomplete inactivation has been discussed above in Reply 3. For measuring recovery from inactivation as presented in Fig. 1c, we have also increased the conditioning pre-pulse time to 30 s, the resulting recovery curve shows two kinetic components of $T_{fast} = 20.1 \pm 3.7$ ms, $T_{slow} = 8714 \pm 2505$ ms, presumably corresponding to the recovery from the residual fast inactivation and slow inactivation, respectively.

New Figure 1: Functional characterization of NavEh^{ΔN}. **a-b.** Representative current traces of NavEh^{WT} (a) and NavEh^{ΔN} (b). A family of currents were conducted by NavEh^{WT} and NavEh^{ΔN} in response to indicated recording protocols. A schematic diagram of the recording protocol is presented on the top of each current trace. **c.** Representative current traces of NavEh^{ΔN} inactivation. A schematic diagram of the recording protocol is presented on the top of each current trace. **d.** Representative current traces of NavEh^{ΔN} recovery from inactivation. Currents were elicited by a pre-pulse at -20 mV for 30 s, followed by an inter-pulse ranging from 1 ms to 33 s at -150 mV. After inter-pulse, a 100 ms test pulse at -20 mV was applied. A schematic diagram of the recording protocol is presented on the top of each current trace. **e.** I-V relationship for NavEh^{ΔN} (Red) and NavEh^{WT} (Black). Currents were normalized to cell capacitance. **f.** Normalized conductance-voltage (G/V) relationship and steady-state inactivation of NavEh^{ΔN} (Red and Blue) and NavEh^{WT} (Black). For measuring G/V curve, currents were elicited by 100 ms depolarizing pulses between -120 mV and 0 mV in step of 10 mV from a holding potential of -150 mV. For measuring steady-state fast inactivation of NavEh^{WT} (black), currents were elicited by a pre-pulse between -140 mV and 0 mV in 5 mV increments for 500 ms and followed by a 50 ms test pulse at -50 mV. For measuring steady-state inactivation of NavEh^{ΔN} (Blue), currents were elicited by a pre-pulse between -120 mV and 0 mV in 10 mV increments for 30 s and followed by a 100 ms test pulse at -20 mV. The Boltzmann fitted data yielded NavEh^{ΔN} activation (Red) $V_{1/2} = -60.7 \pm 1.1$ mV ($n = 5$) and steady-state inactivation (Blue) $V_{1/2} = -78.9 \pm 2.0$ mV ($n = 5$). The NavEh^{WT} (Black) activation $V_{1/2} = -61.5 \pm 2.1$ mV ($n = 15$) and steady-state fast inactivation $V_{1/2} = -94.4 \pm 2.1$ mV ($n = 9$) are adapted from our previous study. **g.** The time course for recovery from steady-state slow inactivation of NavEh^{ΔN}. Recovery curve from slow inactivation was fitted using a double exponential function which yielded two kinetic components $\tau_{fast} = 20.1 \pm 3.7$ ms, $\tau_{slow} = 8714 \pm 2505$ ms ($n = 5$). For each point in this figure, data are means \pm SEM.

5. Is the structure of NavEhWT taken from a previous manuscript? If so, that should be

explicitly defined in the text.

Reply: The two NavEh^{WT} structures (with Ca²⁺ or EGTA) are not from our previous study, which were determined to investigate the previously suggested possible Ca²⁺ dependent activation of NavEh (PMID: 33004614). These two structures have also been deposited to EMDB and PDB databases (See **Supplementary Table 1**). The structures of NavEh^{Ca2+} and NavEh^{EGTA} are almost identical to the structure of our previous study (RMSD = 0.5 Å over 1170 Cα atoms). We have clarified this in the manuscript.

6. The authors suggest the structure of NavEhdN corresponds to a slow inactivated state. However, the electrophysiological data suggests that slow inactivation is incomplete. Are there alternative low-abundance states that were discarded during classification? Additionally, the described slow inactivation process is composed of two components. To which of these phases does the resolved structure correspond and how was such an assignment made?

Reply: We thank Reviewer 3's suggestion on this. To identify any possible conformational changes in this NavEh^{ΔN} structure, we carefully processed the EM data using C1 symmetry for 3D classification. As shown in **Response Figure 6** below, all 3D classes with discernable EM density for the TMs exhibited almost identical conformation, which suggests there is only one dominant state in the EM data. As for the two components of inactivation, we have clarified above that the residual fast inactivation component is most likely mediated by the residual N-terminal fast inactivation particle of NavEh^{ΔN} (**Response Figure 5**). We had shown that the fast inactivation of NavEh is mediated by its N-helix which requires open activation gate to accommodate this fast inactivation particle (PMID: 35581266). Thus, the new structure of NavEh^{ΔN} with apparently smaller activation gate and dilated SF represents a potential slow-inactivated state.

Response Figure 6: 3D classification of NavEh^{ΔN}. **a**, EM reconstruction maps of 3D classification with C1 symmetry imposed. The dashed red square indicates Class3 with the highest resolution which was selected for further refinement. **b**, Superposition of the six 3D class averages from panel a. The black square highlights the activation gates, which are essentially identical among the 6 classes.

7. The selectivity filter of NavEhdN adopts a distorted state, which based on comparisons with potassium channels, leads the authors to propose that the selectivity filter contributes to the inactivated state. The changes in the selectivity filter are subtle. At the moderate resolution of the structure, it would be helpful to include figures in the main text showing the raw density for the selectivity filter to aid the readers in understanding the changes. Especially when calling into significance a movement of 0.2 Å in each monomer for the backbone oxygen of G304.

Reply: We thank Reviewer 3's suggestion on this. We have added EM densities for the SF in the **new Figure 4a,b,d,e** in the revised manuscript (See below). The conformational changes in the SF of NavEh^{ΔN} are quite significant in comparison to that of NavEh^{WT}. Not only the subtle constriction at G304 but also the obvious conformational shifts of E305 can be observed (**Figure 4c** and **f**). Based on this, we speculate that the four enlarged E305 cannot serve as the high-field strength (HFS) site to properly attract and coordinate Na⁺; meanwhile, the constricted SF could prevent extracellular hydrated ions from accessing the SF, thus resulting in the non-conductive inactivated channel. Indeed, the conformational changes in the SF were observed in potassium channels which were proposed to be involved in the C-type inactivation (PMID: 20613835; 35302848). Furthermore, previous mutagenesis studies have shown that the SFs of Nav channels are involved in Nav channel inactivation (PMID: 11892790). For instance, the mutation W402C in the SF of rat Nav1.4 domain I (corresponding to W307 in the SF of NavEh) markedly attenuates slow inactivation (PMID: 8842002); Nav1.5 exhibits more resistance to slow inactivation than Nav1.4, the substitution of V754 (corresponding to L298 in the SF of NavEh) in Nav1.4 DII-SF region with an Ile at the corresponding position of Nav1.5 enhanced the resistance to slow inactivation of Nav1.4 similar to Nav1.5 (PMID: 11325725). These results strongly indicate that the SF regions are involved in the slow inactivation process and could potentially serve as a gate for slow inactivation, which is consistent with our structural observation.

New Figure 4. The dilated selectivity filter of NavEh^{ΔN}. **a-b.** The selectivity filter of Nav_vEh^{WT} (a) and Nav_vEh^{ΔN} (b). Black dashed lines represent the distances within the SFs. EM densities for P2 helices are presented in white surface contoured at 10 σ . **c.** The superposition of the pore region of Nav_vEh^{WT} (gray) and Nav_vEh^{ΔN} (red). E305 is shown side chain in sticks. The red arrow indicates the shifts of E305 between the two SFs. **d-e.** The top view of the selectivity filter of Nav_vEh^{WT} (d) and Nav_vEh^{ΔN} (e). The red and black dashed square represents the SF size of Nav_vEh^{WT} and Nav_vEh^{ΔN}, respectively. The green dashed lines represent the hydrogen-bond between T303 and W307. EM densities for the SFs are presented in white surface contoured at 6 σ . **f.** The superposition of the SF of Nav_vEh^{WT} (gray) and Nav_vEh^{ΔN} (red). The red and green arrows indicate the shifts of E305 and W307, respectively. **g.** The top view of the four-fold symmetric SF of Nav_vAb^{I217C} (PDB code: 3RVY). The black dashed square represents the SF size of Nav_vAb^{I217C}. **h.** The top view of the two-fold symmetric SF of Nav_vAb^{WT} (PDB code: 4EKW). The opposing two subunits are colored in white and green, respectively. The black and red dashed lines represent the SF size of Nav_vAb^{WT}. **i.** The superposition of the SFs of Nav_vAb^{I217C} (light purple) and Nav_vAb^{WT} (gray). E177 residues are shown side-chain in sticks.

8. Continuing from point 7, it is unclear that the distortions described in the selectivity filter would prevent ion permeation. A much more thorough evaluation of the effect of these changes on ion permeation should be performed as the selectivity filters of potassium channels are quite distinct from those of sodium channels and it is not clear that the distortions would lead to similar functional effects.

Reply: We thank Reviewer 3's comments on the SF differences between Nav channels and potassium channels. The distortions were observed in the SFs of potassium channels

and NavEh^{ΔN}, which play a similar but not identical role in slow inactivation. Potassium channels possess a conserved narrow and long SF, which preferentially allows dehydrated K⁺ ions flowing through via coordination with the main-chain carboxyl oxygen atoms (PMID: 11689935); in contrast, the SFs of Na_v and Ca_v channels are wider and shorter than that of potassium channels (PMID: 21743477; 24270805; 31766050; 31866066), which was suggested to form a high-field strength (HFS; E305 in Na_vEh) site above the narrowest site of the SF to attract and dehydrate Na⁺ or Ca²⁺, then the partially dehydrated ions passing through the short SF. The distortions in the SF of potassium channel directly disrupt the coordination and transferring of K⁺; while the dilation of E305 in NavEh^{ΔN} leads to an impaired HFS site which may not be able to coordinate and partially de-hydrate extracellular Na⁺. In addition, the slightly constricted SF of NavEh^{ΔN} (at G304) would prevent fully-hydrated ions from passing through the SF.

9. The timescales for the closure of the activation gate are much faster than would be expected for slow inactivation. In the replicates presented in Figure 6, all of the activation gates collapse within 700 ns, whereas currents can be measured from channels in cells after 10 seconds. The collapse in the simulations thus does not appear to correspond to the slow inactivation seen in electrophysiological recordings.

Reply: We thank We appreciate Reviewer 3's comments on the MD simulations, which was also pointed out by Reviewer 1. While we did perform MD studies on NavEh^{ΔN} as suggested by Reviewer 1, we realized that the duration of our simulations, limited to the microsecond timescale, may not be sufficient to fully capture the desired conformational changes between the open and slow-inactivated states, which may occur over a longer timescale of several seconds. Consequently, the results obtained from these MD studies may not sufficiently explain the observations from the experiments. In light of these considerations, we have made the decision to exclude the MD-related sections from the revised manuscript.

The rationale behind this decision together with our next-step plan are detailed as follows:

Initially, we aimed to use MD simulations to investigate the conformational changes between the open state (represented by NavEh^{WT} structure with N-helix removed) and the slow-inactivated state (represented by NavEh^{ΔN} structure). Six independent MD trajectories, each lasting 3-4 microseconds, were performed on the pore domains of the NavEh^{WT} and NavEh^{ΔN} structures.

After analysis of the simulations, we did observe closed activation gate from both simulations which are consistent with previous E-phys results and MD simulation results of mammalian Nav1.5 (PMID: 34520724; 7876824); but unfortunately, we did not observe any significant conformational changes in the P2 helix within the selectivity filter (**Response Figure 7**). Additionally, the observed changes in the channel radii of active state could not explain the experimental observations (Figure 6a in the previous version of manuscript).

Response Figure 7. P2 helix conformation dynamics during MD study. The distance between Ca atom of A310 and Ca atom of P314 is used to indicate the conformation of P2 helix. When P2 helix adopts a standard alpha helix conformation in NavEh^{WT}, the Ca distance between A310 and P314 is around 7 Å. When P2 helix adopts a helix-loop-helix conformation in NavEh^{ΔN}, the Ca distance between A310 and P314 is around 11 Å.

We speculate that two factors may have contributed to the inability of our MD study to establish a connection between the open and slow-inactivated states. The first factor is the limited timescale of our sampling strategy, which was constrained to microsecond timescales. In contrast, based on experimental findings, the conformational changes between the states may require a timescale of seconds from the electrophysiological experiments. To address this limitation, we are actively exploring enhanced sampling methods, such as meta-dynamics. The second factor is the truncation of the voltage-sensing domain (VSD) during our MD simulations. It is possible that this truncation impacted the integrity of the whole system and thereby block the desired conformational changes. However, including the four VSDs in our current sampling process would significantly increase the computational resources required. Nevertheless, we anticipate that future MD simulations incorporating the VSD and employing meta-dynamics might provide valuable insights into the dynamics in the conformational changes between the open and slow-inactivated states, as well as the role of the VSD in this process.

Considering these limitations, we have opted to remove the MD contents from the current manuscript. This decision will allow us to dedicate more time and employ new sampling methods to conduct further MD studies on the revised systems including the VSDs. We aim to publish these results in future studies, which will further enhance our understanding of the system.

10. Altogether, the functional state to which the determined structure corresponds is not clear from the analyses provided. It is essential that the authors uncover the functional state of this structure and elucidate the roles of the selectivity filter and activation gate in slow inactivation.

Reply: With all the helpful suggestions from all reviewers, we believe that the revised manuscript has been substantially improved. We have clarified that the residual fast inactivation component of NavEh^{ΔN} is most likely caused by the N-terminal residual fast

inactivation particle and showed that NavEh^{ΔN} indeed underwent slow inactivation which is significantly distinct from the fast inactivation of NavEh^{WT}. We further showed that the functional properties of NavEh^{ΔN} is highly similar to that of human Nav1.5 F1485Q mutant, a channel is known to have slow inactivation. These findings also suggest the interactions between slow inactivation gate and fast inactivation gate. We speculate that the communication between fast and slow inactivations is mediated by the fast inactivation particles interacting with the intracellular activation gate. The structure of NavEh^{ΔN} features activated VSDs, a dilated non-conductive SF, and a non-conductive activation gate, assuming a potential slow-inactivated state that is distinct from previous Nav structures. This possible slow-inactivated structure highlights that the SF and the activation gate could both serve as gates for slow inactivation.

Minor Concerns:

1. Figure 1d – Y282 appears to be a threonine

Reply: Thanks. Corrected.

2. What is the red dashed box in Figure 3d-e?

Reply: The red dashed box in Figure 3d-e represents the region of activation gate. We have clarified this in the figure legend of the revised manuscript.

Reviewer #1 (Remarks to the Author):

I read the answers to my and other reviewers' remarks and I think the paper is improved with the new functional and mutagenesis studies and discussion. However, I disagree with the complete removal of the MD section. My comment was intended to test if the cryo-obtained structure is stable (within the limitations of MD, of course). Although I agree that the conformational transition occurs on longer time-scales, and studying it is a challenging task worth to be pursued afterwards, extensive standard MD of the new cryo structure can support its validity. Is the structure stable (meaning, are its main features preserved over the MD time-scale, as for example the dilated SF and fenestrations)? How do distances between opposing helices behave over time? What are the mostly visited SF conformations? I suggest to thoroughly analyze the already performed simulations, specifically highlighting the differences between the delta-WT and the new cryo, in order to further support the reliability of the latter.

Reviewer #3 (Remarks to the Author):

The authors have substantially improved the manuscript and it is now suitable for publication.

Response to Reviewers' Comments

Reviewers' Comments:

Reviewer #1 (Remarks to the Author):

Reviewer #1 (Remarks to the Author):

I read the answers to my and other reviewers' remarks and I think the paper is improved with the new functional and mutagenesis studies and discussion. However, I disagree with the complete removal of the MD section. My comment was intended to test if the cryo-obtained structure is stable (within the limitations of MD, of course). Although I agree that the conformational transition occurs on longer time-scales, and studying it is a challenging task worth to be pursued afterwards, extensive standard MD of the new cryo structure can support its validity. Is the structure stable (meaning, are its main features preserved over the MD time-scale, as for example the dilated SF and fenestrations)? How do distances between opposing helices behave over time? What are the mostly visited SF conformations? I suggest to thoroughly analyze the already performed simulations, specifically highlighting the differences between the delta-WT and the new cryo, in order to further support the reliability of the latter.

Reply: We thank Reviewer 1 for his/her positive comments on the improvement of our manuscript and suggestions on thoroughly analyzing the MD results. As suggested, we have performed additional MD simulations on the pore of NavEh^{WT} with the N-helix, and we have carefully analyzed the MD results of NavEh^{WT} (without the N-helix), NavEh^{WT} (with the N-helix), and NavEh^{ΔN}. We have revised our manuscript mainly from two perspectives: 1. Including MD analysis for the validation of the cryo-EM structures of NavEh^{WT} and NavEh^{ΔN}. To do this, we conducted an additional molecular dynamics (MD) simulation on the pore domain of NavEh^{WT} (new Supplementary Fig. 10a). The reason of using newly performed NavEh^{WT} MD system instead of previously performed NavEh^{WT} without the N-helix is that the cryo-EM structure of NavEh^{WT} contains the N-helix, while the using of NavEh^{WT} without the N-helix would not align with our purpose of validating the cryo-EM structures. Below paragraph was added to the revised manuscript (Lines 264 to 277):

“To validate the cryo-EM structures of NavEh^{WT} and NavEh^{ΔN}, we conducted molecular dynamics (MD) simulations using the pore domains of NavEh^{WT} and NavEh^{ΔN} as initial configurations for a duration of 2 μs (Supplementary Fig. 10a). We investigated the stability of key structural features, including the dilation of the SF, size of fenestrations, and distances between opposing helices. Specifically, we analyzed the distance between the alpha carbon (Ca) atoms of residues A310 and P314, the distance between the side chain centroids of F299 and F218, and the pore radius as indicators of SF dilation, fenestrations, and distances between opposing helices, respectively. The consistent values of these indicators suggest the stability of these key structural features (Supplementary Fig. 10b-d). Furthermore, a clustering analysis was performed on the MD results to identify the most frequently observed conformations of the SF, which shows two major clusters (comprising 85% of the population) for the SF of NavEh^{ΔN} and seven major clusters (each comprising over 5% of the population) for the SF of NavEh^{WT} (Supplementary Fig. 10e).”

Supplementary Figure 10. MD simulations of the pore domain of NavEh^{WT} and NavEh^{ΔN}.

a. The protein backbone RMSD plots for each simulation system. The protein structures in each trajectory were aligned with the initial structure using the Least Squares algorithm before computing the RMSD. **b.** Dynamics of pore radius. For NavEh^{WT}, the N helix was included during the simulation, but it was removed for the calculation of the pore radius using the HOLE program. **c.** Dynamics of distance between Ca atoms of A310 and P314. This distance was utilized as an indicator of the dilation of the selectivity filter. **d.** Dynamics of distance between side centroids of F299 and F218. This distance served as a measure of the fenestration in the protein structure. In panel c and d, when the distance of interest involved four values due to the symmetry of the protein, the minimum of the four values was used for the plot. **e.** Cluster analysis of 2000 ns MD simulation for each system. The major clusters accounting for over 5% of the population were indicated with red boxes and the conformations of the selectivity filter from representative structures from these clusters were displayed.

2. Including MD analysis as a prelude to the future research direction based on our study. To do this, we included the previously analysis on NavEh^{WT} without the N-helix, and added

the below paragraph to the section (Lines 278 to 302):

“To further dissect the relationship between activated and inactivated states of NavEh, we performed a 4 μ s MD simulation on the open pore domain of NavEh^{WT} structure by removing the N-helix. After plotting the pore radius at the activation gate of each MD simulation trajectory, the results show that the gate remained open for a duration of 300-680 ns and subsequently became closed (Supplementary Fig. 11a). Consistent with the findings of previous studies^{14,52}, this result confirms that the open gate of Nav channels is inherently unstable. The clustering of the MD trajectories with a RMSD cutoff of 1.5 Å resulted in 4-6 populations for each simulation, of which the majority populations displayed closed activation gates (Supplementary Fig. 11b and 11c). We selected the representative trajectory of the most populated cluster of the five MD simulations, and compared them with the pore domain of NavEh^{ΔN}. The superposition reveals that the MD representatives are highly similar to the EM structure with backbone RMSD of 1.5-1.8 Å, especially between the activation gates (Supplementary Fig. 11d). Notably, the P2 helix of the five MD representatives adopt an intact α -helical conformation similar to that of NavEh^{WT}, unlike the broken P2 helix in NavEh^{ΔN} (Supplementary Fig. 11e and 11f). This observation suggests that under our simulation conditions, it is difficult to achieve the slow-inactivated state, in line with the observation that the development of slow inactivation on the order of tens of seconds^{53,54}. To tackle this limitation, we are actively exploring enhanced sampling methods, such as meta-dynamics. The second factor is the truncation of the VSD during our MD simulations. It is possible that this truncation would impair the integrity of the whole system and thereby blocks the desired conformational changes. Nevertheless, we anticipate that future MD simulations incorporating the VSD and employing meta-dynamics will provide valuable insights into the role of the VSD in the conformational changes between the active and inactivation states of Nav channels.”

Supplementary Figure 11. MD simulations of NavEh pore domain without the N-helix.

a. Pore radius fluctuations in the five independent MD simulation trajectories of NavEh pore domain without the N-helix. T represents Trajectory, and the time point at which the gate closed is indicated with a red arrow. The structures of closed activation gate from MD are displayed as cartoons. The visualized duration here is limited to 1,200 ns. **b.** Clustering analysis on full-length trajectories of the NavEh pore domain without the N-helix. The ion path of the representative structure from the most populated cluster is depicted for each trajectory. In the bar chart, the closed gate is represented by a red "C" and the open gate is represented by a green "O". **c.** The pore radii of the representative structures from the most populated cluster of each trajectory for NavEh pore domain without the N-helix. **d.** The superposition of the five representative structures from the most populated cluster of each trajectory and the NavEh^{ΔN} pore (red). **e** and **f.** Pore-loop comparison between and the five representative structures of the MD simulations from panel d with NavEh^{WT} (**e**) and NavEh^{ΔN} (**f**). Red arrow indicates the conformational difference between the P2 helices.

Reviewer #3 (Remarks to the Author):

The authors have substantially improved the manuscript and it is now suitable for publication.

Reply: Thank you for your support for the publication of this work.

Reviewer #1 (Remarks to the Author):

The additions are quite interesting and consistent with the other findings, I strongly support publication. Axis labels should be added to Figure S10 panels a-d (no need for another round of review).

Response to Reviewers' Comments

Reviewers' Comments:

Reviewer #1 (Remarks to the Author):

Reviewer #1 (Remarks to the Author):

The additions are quite interesting and consistent with the other findings, I strongly support publication. Axis labels should be added to Figure S10 panels a-d (no need for another round of review).

Reply: We thank Reviewer 1 for his/her support of publication of our study. We have added the labels for the axis in the Supplementary Figure 10 in the revised manuscript.